# Recurrent network interactions explain tectal response variability and experience-dependent behavior

**Asaph Zylbertal\*, Isaac H Bianco\***

Department of Neuroscience, Physiology & Pharmacology, University College London, London, United Kingdom

**Abstract** Response variability is an essential and universal feature of sensory processing and behavior. It arises from fluctuations in the internal state of the brain, which modulate how sensory information is represented and transformed to guide behavioral actions. In part, brain state is shaped by recent network activity, fed back through recurrent connections to modulate neuronal excitability. However, the degree to which these interactions influence response variability and the spatial and temporal scales across which they operate, are poorly understood. Here, we combined population recordings and modeling to gain insights into how neuronal activity modulates network state and thereby impacts visually evoked activity and behavior. First, we performed cellular-resolution calcium imaging of the optic tectum to monitor ongoing activity, the pattern of which is both a cause and consequence of changes in network state. We developed a minimal network model incorporating fast, short range, recurrent excitation and long-lasting, activity-dependent suppression that reproduced a hallmark property of tectal activity – intermittent bursting. We next used the model to estimate the excitability state of tectal neurons based on recent activity history and found that this explained a portion of the trial-to-trial variability in visually evoked responses, as well as spatially selective response adaptation. Moreover, these dynamics also predicted behavioral trends such as selective habituation of visually evoked prey-catching. Overall, we demonstrate that a simple recurrent interaction motif can be used to estimate the effect of activity upon the incidental state of a neural network and account for experience-dependent effects on sensory encoding and visually guided behavior.

**\*For correspondence:**
a.zylbertal@ucl.ac.uk (AZ);
i.bianco@ucl.ac.uk (IHB)

**Competing interest:** The authors declare that no competing interests exist.

## Editor's evaluation

This important study investigates how neural activity states contribute to and shape sensory responses using a combination of neuronal activity imaging and computational modeling. They show that recurrent connectivity in networks can shape sensory responses in an experience-dependent manner and can be used to explain variability in experimentally-observed neuronal responses to sensory stimuli.

## Introduction

Neural and behavioral responses to sensory input are influenced both by stimulus properties as well as the internal state of the brain, enabling animals to generate flexible, context-dependent behavior. Consequently, repeated presentation of identical sensory cues typically evokes variable activity and this variation is left unexplained by analyses restricted to response averaging, as is traditionally used for empirically estimating neuronal tuning properties. Increasingly, however, such traditional approaches are being replaced by treating trial-to-trial variability as a product of interactions between the external

input and the incidental state of the brain (*Churchland et al., 2011*; *Morcos and Harvey, 2016*; *Urai et al., 2019*).

Internal state is multidimensional and influenced by many factors including activity of neuromodulatory populations (*Nicola et al., 2003*; *Marques et al., 2019*; *Mu et al., 2019*; *McCormick et al., 2020*) and short- and long-term changes in synaptic strength (*Magee and Grienberger, 2020*). At the single neuron level, it can be summarized as fluctuations in excitability (the degree to which a cell is activated by a given input; *Arieli et al., 1996*; *Stephani et al., 2021*) such that the state of the population is defined by the excitability of its constituent cells. One factor that influences single neuron excitability is recent network activity, fed back by local and/or long-range recurrent connections (*Mohajerani et al., 2013*; *Zylbertal et al., 2017a*; *Smith et al., 2018*). For example, activity may briefly shift the excitability of post-synaptic neurons as a result of direct synaptic transmission, or exert longer-term changes through secondary processes such as vesicle release dynamics (*Shu et al., 2006*) or modulation of intrinsic properties (e.g. ion channel dynamics; *Zhang and Sillar, 2012*; *Zylbertal et al., 2015*). By and large, changes in excitability cannot be measured directly and so this component of brain state remains hidden to experimental examination. As a result, the extent to which neural activity modulates network state, the spatial and temporal scales of such interactions, and their impact on sensorimotor processing are unclear. Here, we set out to estimate fundamental properties of these interactions and test the hypothesis that activity-dependent variations in network state explain trial-to-trial response variability and ultimately behavior.

Ongoing activity, persisting in the absence of changes in the sensory environment, provides a window into the interactions between activity and network state. Such activity is commonplace in sensorimotor brain regions, including multiple cortical areas (*Kenet et al., 2003*; *Yuste et al., 2005*), and the optic tectum (OT; *Romano et al., 2015*). Ongoing activity is, on the one hand, a cause of fluctuations in network state, as evidenced by the variability it introduces to sensory responses (*Arieli et al., 1996*; *Petersen et al., 2003*; *He, 2013*; *Shimaoka et al., 2019*). Conversely, patterns of ongoing activity are also a consequence of brain state dynamics and may be altered for extended periods of time by transient inputs (*Fiser et al., 2004*; *Abbott et al., 2009*; *Deneux and Grinvald, 2016*; *Chen and Gong, 2019*; *Franco and Yaksi, 2021*; *Fritsche et al., 2021*). This bidirectional nature, along with the observations that ongoing activity typically displays a characteristic correlation structure between individual neurons and across time (*Tsodyks et al., 1999*; *Kenet et al., 2003*; *Smith and Kohn, 2008*; *Smith et al., 2018*) and recapitulates some features of sensory-evoked activity (*Luczak et al., 2009*), makes it an attractive handle for studying the spatiotemporal relationships between neural activity and network state.

Our strategy, therefore, was first to use light-sheet calcium imaging to observe ongoing activity in the complete optic tectum of larval zebrafish and develop a minimal network model that could reproduce the core features of these dynamics based on defined interactions between neural activity and excitability. In the model, every neuron is subject to the same recurrent circuit interactions comprising fast, local excitation and slower, long-range suppression. Next, we were able to use the model as a tool to estimate the activity-dependent state of the biological network. Specifically, we estimated the excitability of every neuron in OT based on the recent history of observed spiking and found this model-derived estimate of network state explained a portion of the trial-to-trial variability in responses to visual prey-like stimuli. The model also successfully predicted the reciprocal effect, where stimulus-evoked activity had a lasting effect upon network state producing prolonged, spatially specific suppression of both ongoing and visually evoked responses. These model-estimated changes in brain state were also associated with modulation of both spontaneous and visually evoked prey-catching behavior. In sum, a recurrent interaction motif was able to explain, in part, how activity influences the incidental state of the tectal network and thereby contributes to variability in visual representations and behavior.

## Results

### The spatiotemporal structure of ongoing activity in the optic tectum

Ongoing activity both arises from and contributes to fluctuations in neuronal excitability. To gain insight into the dominant network interactions that mediate these processes, we analysed ongoing activity in the optic tectum (OT). To measure this activity, we performed light-sheet calcium imaging

of transgenic larvae expressing GCaMP6s (elavl3:H2B-GCaMP6s, 6 dpf, n=14), initially under constant environmental conditions (i.e. without presenting visual stimuli). We imaged a 375x410 × 75 µm volume encompassing OT and adjacent brain structures, at a rate of 5 volumes per second (*Figure 1A–B*). Following motion correction, each imaging plane was automatically segmented to individual neurons (*Kawashima et al., 2016*), and spikes were inferred by deconvolution of the extracted fluorescence signal (*Friedrich et al., 2017*). Imaged volumes were registered to a reference brain (*Avants et al., 2009*; *Marquart et al., 2015*) and somata in the stratum periventriculare (SPV), which comprise the majority of tectal neurons, were considered for further analysis (*Figure 1B*, $1.4 \cdot 10^4 \pm 3 \cdot 10^3$ tectal SPV cells per fish, mean ± SD).

The spatial pattern of activity correlations was compatible with recurrent excitatory interactions between tectal cells. To show this, we analyzed the Pearson correlation between the ongoing activity of each neuron ('seed'), and the activity of all other neurons during the entire experimental session. Correlation coefficients smoothly and monotonically decreased as a function of the Euclidean distance between neurons (*Figure 1C* and *Figure 1—figure supplement 1A–C*). We confirmed this finding using two-photon imaging, suggesting it is not a consequence of light scattering in the light-sheet microscope but rather a feature of tectal activity (*Figure 1—figure supplement 1C*). We projected neuronal coordinates into two dimensions (anterior-posterior and medial-lateral tectal axes, see Materials and methods), producing a 2D correlation map for each seed neuron (*Figure 1D*). A 2D Gaussian was fitted to each map, excluding the seed neuron itself, yielding good fits for most cells ($r^2$=0.42 ± 0.17, mean ± SD, *Figure 1D* and *Figure 1—figure supplement 1A–C*). The peaks of the fitted Gaussians showed close correspondence to the locations of seed neurons (*Figure 1E* and *Figure 1—figure supplement 1A–C*). Comparably good fits were obtained by fitting an exponential decay function ($r^2$=0.39 ± 0.18, mean ± SD), and again the peaks of fitted exponentials corresponded to the locations of seed neurons (*Figure 1—figure supplement 1A–C*). These data are compatible with a recurrent network in which excitatory interactions smoothly decay with distance from each cell.

A salient aspect of tectal activity is the occurrence of 'spontaneous' bursts, during which spatially compact groups of neurons briefly fire in synchrony (*Bianco and Engert, 2015*; *Romano et al., 2015*; *Avitan et al., 2017*). To detect bursts, we evaluated the spatial distribution of cells active during one-second windows centred on population activity peaks (*Figure 1F*). For each window, we identified the cells participating in a burst using density-based DBSCAN clustering (*Ester et al., 1996*; *Figure 1—figure supplement 1*, Materials and methods). The temporal extent of each burst was then defined as the period of non-zero Poisson-filtered activity of participating neurons (*Figure 1F*, iv). Overall, 54,424 bursts were identified in 14 fish. On average, each activity peak was associated with 1.5±2.0 localised bursts (mean ± SD), bursts occurred at a rate of 46±11 per minute and appeared uniformly distributed in space and time (*Figure 1G*). Bursts were brisk, phasic events, with the majority of spiking occurring over a timecourse of around one second (*Figure 1H*).

The statistics of bursting suggested tectal activity is shaped by a second interaction, where activity results in prolonged reduction in the excitability of tectal cells. As has been previously reported, the sizes and durations of bursts were highly variable (95±183 neurons per burst, 2.5±3 s duration, mean ± SD), and distributed according to a power law (*Figure 1I–J*), suggestive of a critical phenomenon with 'avalanche' dynamics (*Ponce-Alvarez et al., 2018*). A universal feature in such phenomena is a self-limiting element responsible for both event termination and suppression of subsequent events. Therefore, we analysed how participating in a burst influences subsequent spiking activity (*Figure 1K*) and burst participation probability (*Figure 1L*) for individual neurons. Cells were less active and less likely to take part in another burst for tens of seconds. These observations are compatible with activity-dependent suppression that has a self-limiting effect on bursting as well as a lasting effect extending beyond burst termination.

In sum, ongoing tectal activity is compatible with two types of intercellular interaction that might shape network dynamics. First, activity correlation is suggestive of excitatory recurrent connectivity that smoothly decays with distance. Second, the dynamics of bursting suggest prolonged activity-dependent suppression of excitability.

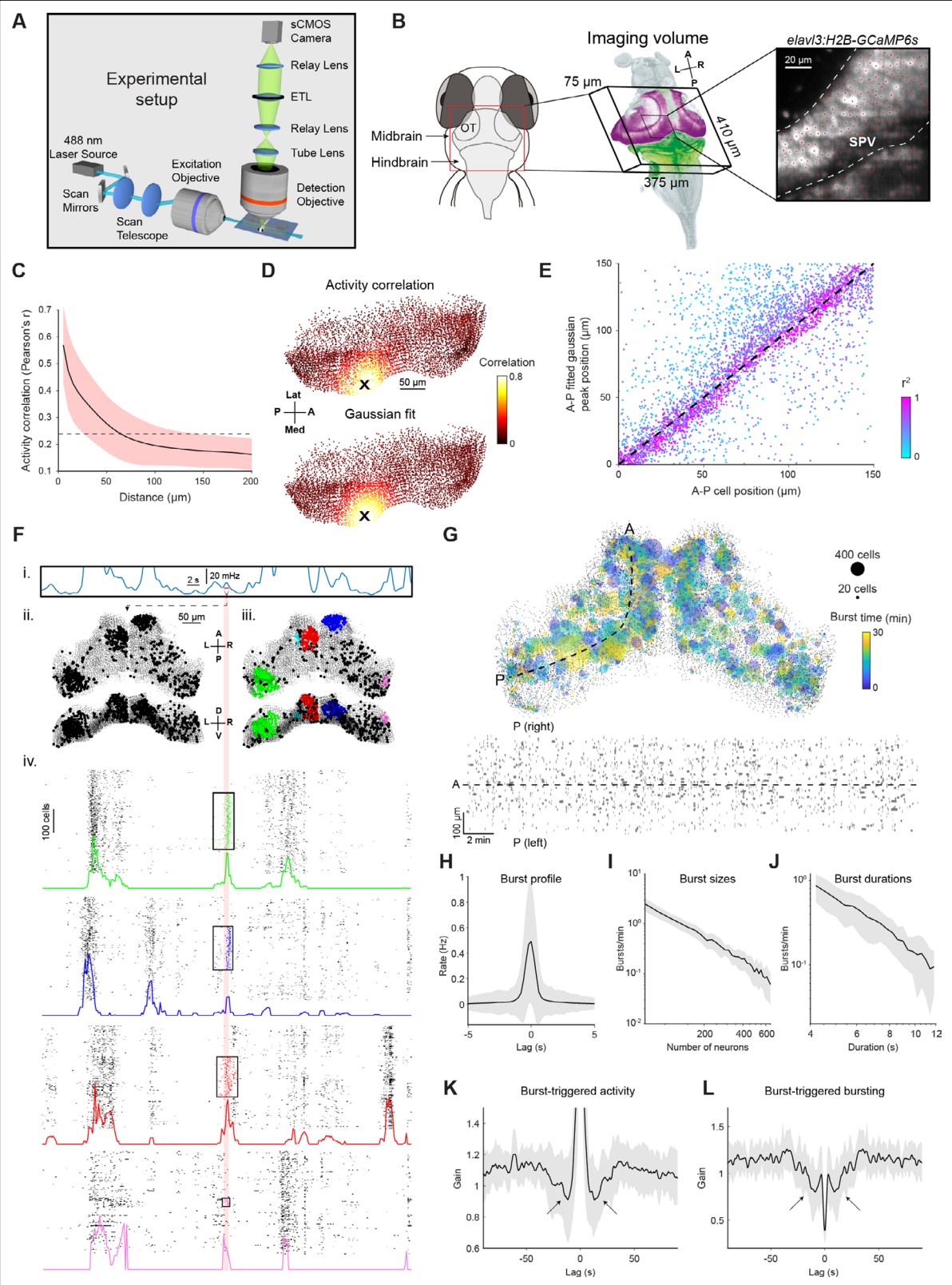

**Figure 1.** Tectal activity displays a uniform spatial correlation structure and localised bursting. (**A**) Light-sheet calcium imaging of tethered larval zebrafish (not to scale). (**B**) Left: Imaging field-of-view (green with tectal SPV mask in magenta) following registration to a reference brain (grey). Right: Section of a single imaging plane showing centroids of identified neurons (red). (**C**) Pairwise activity correlation as a function of Euclidean distance between cells in a single fish. Red shaded area denotes SD across seed cells (n=14597). Dotted line indicates mean correlation. (**D**) Map showing

*Figure 1 continued on next page*

*Figure 1 continued*

Pearson's correlation between the activity of an example seed cell (X) and all other cells (top), and corresponding Gaussian fit (bottom). (**E**) Location of fitted correlation peaks vs location of seed cells in the same fish as D. Color indicates $r^2$ of Gaussian fits. (**F**) Example illustrating burst detection procedure. (i) Population activity peaks detected from mean spiking activity across all tectal neurons. (ii) Active cells identified within a 1 s window centred on an activity peak (red line). (iii) Active cells clustered according to their spatial density. (iv) Burst initiation and termination times determined for each cluster using Poisson-filtered activity of constituent neurons. Raster plots show the clustered cells participating in a burst as well as 200 neighbouring cells. Colored traces show Poisson-filtered activity and rectangles denote the extent of each burst in time and space. (**G**) Top: Locations of all bursts detected during a single 30 min imaging session. Colors indicate burst times and spot sizes indicate number of participating neurons. Dashed line indicates the curved axis used for inferring A-P coordinates in (D-E). Bottom: Locations of burst centroids along A-P axis versus burst times. Rectangle widths indicate duration of each burst. (**H**) Mean activity of burst-participating neurons, centred on the peak of the Poisson-filtered activity. (**I–J**) Histogram of burst sizes (number of participating neurons, I) and durations (J). The rates of both quantities vary according to a power law, resulting in straight lines in a log-log plot. (**K**) Post-burst activity suppression: mean spiking of burst-participating neurons triggered on burst time. Data is normalised by mean activity of each neuron triggered on randomly sampled times. Arrows indicate post-burst activity suppression. (**L**) Probability of participation in a burst, triggered on burst time. Data is normalized for each fish by random circular permutation of burst times. Arrows indicate post-burst suppression of burst participation. Plots in H-L show mean values with grey shaded areas indicating SD across fish (n=14). ETL, electrically focus-tunable lens; OT, optic tectum; A, anterior; P, posterior; L, left; R, right; D, dorsal; V, ventral; Med, medial; Lat, lateral. See also *Figure 1—figure supplement 1*.

The online version of this article includes the following source data and figure supplement(s) for figure 1:

**Source data 1.** Data provided as a MATLAB structure.

**Figure supplement 1.** Ongoing activity and localized bursting in the optic tectum.

## A minimal spiking network model with recurrent interactions reproduces tectal bursting

Next, we used a computational modeling framework with defined interactions between spiking activity and excitability state to explore the possibility that tectal bursting emerges from the recurrent interaction motifs suggested by our analysis of ongoing activity. Theoretical network analysis has proposed that recurrent excitatory connectivity, balanced by long-range inhibition, produces stable, self-sustaining and spatially defined 'blobs' of activity (*Amari, 1977*). We reasoned that by making the inhibition in such a model depend on a slow time-integration of activity, transient bursts would emerge instead of stable blobs and would be followed by prolonged activity suppression.

To test this idea, we used a probabilistic spiking network simulation based on a linear-nonlinear-Poisson (LNP) formalism. This framework has been successfully used to relate activity in neuronal populations to past activity and sensory input (*Pillow et al., 2005*; *Pillow et al., 2008*; *Truccolo et al., 2005*), and its non-deterministic aspect fits the apparent stochastic nature of tectal bursting. In LNP models, the instantaneous firing rate of each neuron ($\lambda$) is determined by an exponentiated 'linear drive', computed as the sum of filtered inputs from all neurons in the network plus optional external inputs; spikes are then emitted by simulating a stochastic Poisson process according to the firing rate (*Figure 2A*, *Pillow et al., 2008*). In our model, the linear drive can be likened to the excitability state of model neurons. Our aim was to test if a network defined by a recurrent interaction motif, applied equally across all neurons, might be sufficient to reproduce the bursting activity observed in OT, rather than fit the connectivity of each neuron. Our model is thus defined by a set of only seven parameters, describing the spatial and temporal aspects of 'excitatory' ('E') and 'inhibitory' ('I') interactions in the network, resulting in increase and decrease in excitability, respectively. The connection weights between all pairs of neurons decay with Euclidian distance according to Gaussian functions with variances $\sigma^{(E)}, \sigma^{(I)}$, scaled by fixed gains $g^{(E)}, g^{(I)}$. Both interactions decay exponentially over time according to time constants $\tau^{(E)}, \tau^{(I)}$. A fixed bias term, μ, is summed with all other inputs to determine the total linear drive (see Materials and methods). The locations of neurons in the simulation were taken from a single fish registered to the reference brain. Notably, given that the same connectivity rule is applied to every tectal cell, the network is fully described by the set of seven parameters and the locations of the neurons.

In accordance with our hypothesis, using this generative model to simulate network activity resulted in spatially localised bursting (*Video 1*). Bursting was a robust emergent property that was observed across a broad range of parameter values (see below). Bursts could be identified by analysing the simulation spiking output using the same method as for experimental data (described above) and were uniformly distributed in space and time with a range of sizes and durations (*Figure 2B–C*).

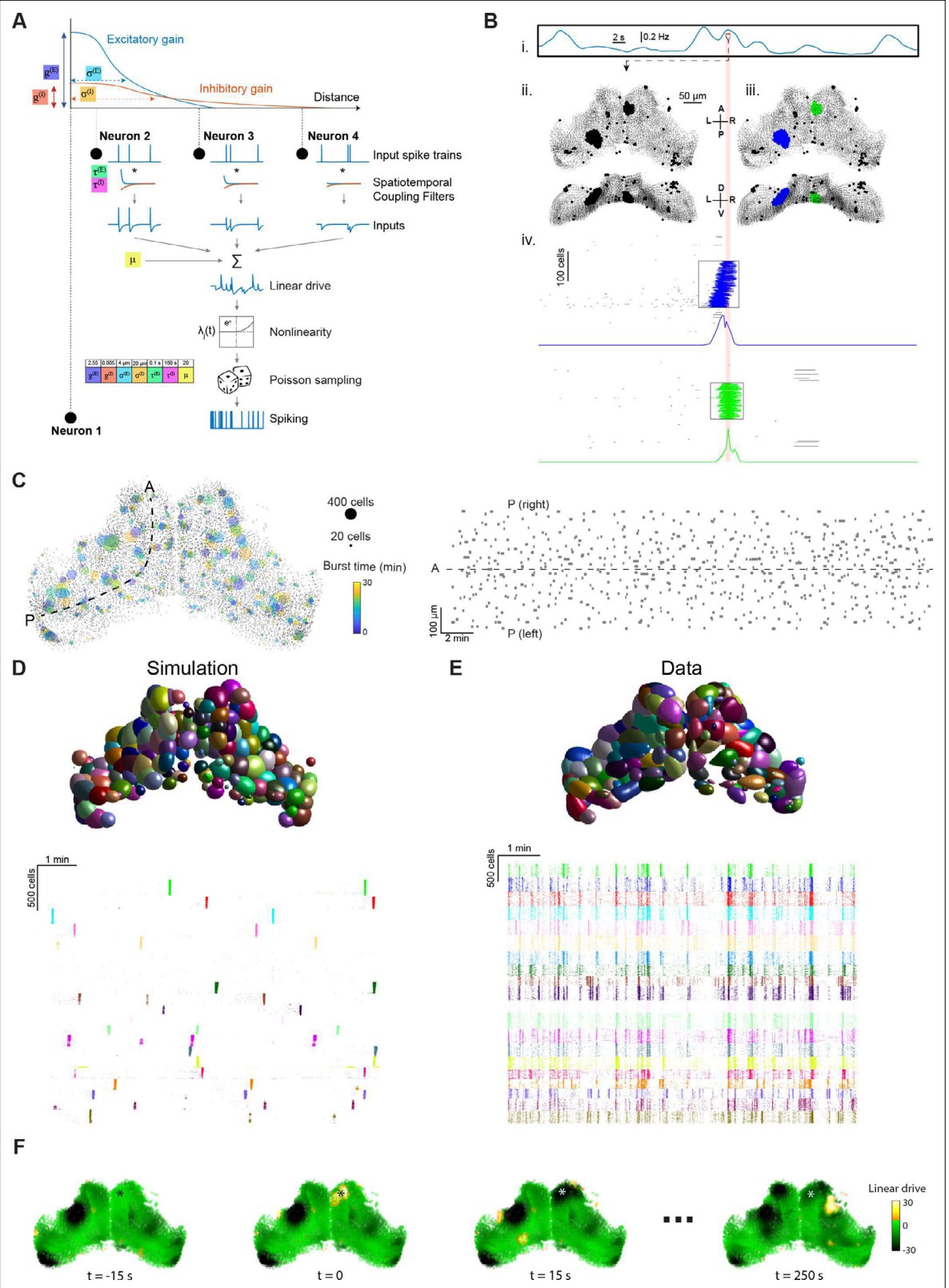

**Figure 2.** Stochastic spiking network model reproduces tectal bursting. (**A**) LNP Model architecture: connection weights for excitatory (E) and inhibitory (I) interactions are determined by Gaussian functions of the intercellular Euclidian distance, with unique gain ($g^{(E)}$, $g^{(I)}$) and spatial standard deviation ($\sigma^{(E)}$, $\sigma^{(I)}$) for each type of interaction. Presynaptic spikes are filtered in time with time constants ($\tau^{(E)}$, $\tau^{(I)}$) and summed along with a bias (μ) to

*Figure 2 continued on next page*

*Figure 2 continued*

produce the linear drive (excitability state). Exponentiating this linear drive sets the mean of an inhomogeneous Poisson process from which spikes are randomly emitted (Materials and methods). Model parameters used for all panels in this figure are shown inset. (**B**) Example of burst detection in the simulation results (c.f. *Figure 1F*). (**C**) Left: Locations of bursts detected during a 30 min simulation. Colors indicate burst times and spot sizes indicate number of participating neurons. Right: Burst locations vs. burst time. Rectangle widths indicate the temporal extent of each burst. (**D–E**) Neuronal assemblies detected using PCA-promax algorithm (Materials and methods), for simulation results (D) and experimental data (E), shown on a spatial map (top) and a raster plot for representative assemblies (bottom). (**F**) Time-course of linear drive to model cells around the time of a spontaneous burst (t=0). Star indicates burst centroid.

The online version of this article includes the following source data for figure 2:

**Source data 1.** Data provided as a MATLAB structure.

In previous studies, localised bursting in the zebrafish OT has been attributed to sub-networks with enhanced connectivity, termed neuronal 'assemblies' (*Romano et al., 2015*; *Avitan et al., 2017*; *Marachlian et al., 2018*; *Mölter et al., 2018*; *Diana et al., 2019*). A PCA-promax algorithm was previously used to identify such assemblies based on the tendency of neurons to be co-active (*Romano et al., 2015*). We applied this method to our simulated tectal network and despite the fact that, by design, there were no subgroups of cells with preferential connectivity, PCA-promax yielded multiple spatially localised 'assemblies' (*Figure 2D*), as was also the case for recorded data (*Figure 2E*). This analysis demonstrates that repeated, synchronised bursting of tectal neurons can occur in a network where every cell has the same distance-based recurrent connectivity and therefore that this feature of activity does not, by itself, imply the existence of preferentially connected subnetworks.

Our simulated tectal network enabled us to examine the linear drive of model neurons — in effect their excitability state. This revealed (*Figure 2F*, *Video 2*) that bursts tend to occur in regions with higher than average excitability (bright green) and leave behind a region of low excitability (black), supressing activity in the same region for a long period of time.

In sum, our network model reveals that a simple recurrent interaction motif is sufficient to explain tectal bursting.

## Model optimization reproduces the statistics of tectal bursting

We next evaluated the capacity of the model to produce bursting with biologically relevant statistics. We found that bursting occurred robustly across a range of parameter values, albeit with burst statistics – such as average burst size and duration – varying across parameter space.

Systematically varying the properties of the inhibitory interaction (space constant $\sigma^{(I)}$, time constant $\tau^{(I)}$ and gain $g^{(I)}$) produced a wide range of mean burst frequencies (0.1–245 bursts per minute) and sizes (18–356 neurons per burst), while some combinations of parameters completely abolished bursting, at least for the 30 min duration tested (*Figure 3A*). For example, eliminating the inhibitory interaction ($g^{(I)} = 0$) abolished bursting (*Figure 3A*, green dashed line) since all neurons remain tonically active without a suppressive

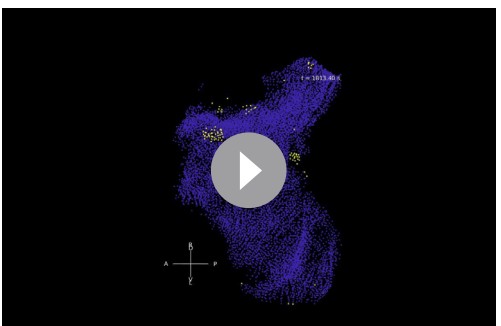

**Video 1.** Spiking network model reproduces localised bursting. Example simulation. Spiking neurons are depicted as yellow dots, neurons participating in a detected burst are marked by fading circles. Quiescent periods have been omitted. A, anterior; P, posterior; L, left; R, right; D, dorsal; V, ventral.
https://elifesciences.org/articles/78381/figures#video1

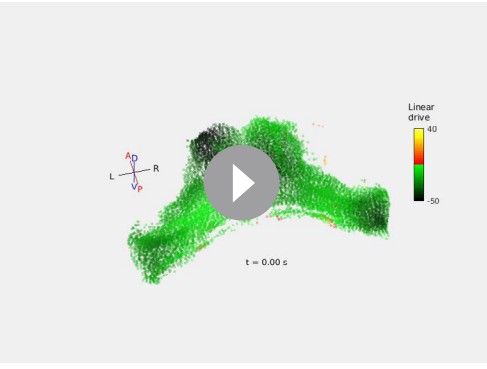

**Video 2.** Simulated linear drive. Time-course of linear drive to model cells in a single simulation run.
https://elifesciences.org/articles/78381/figures#video2

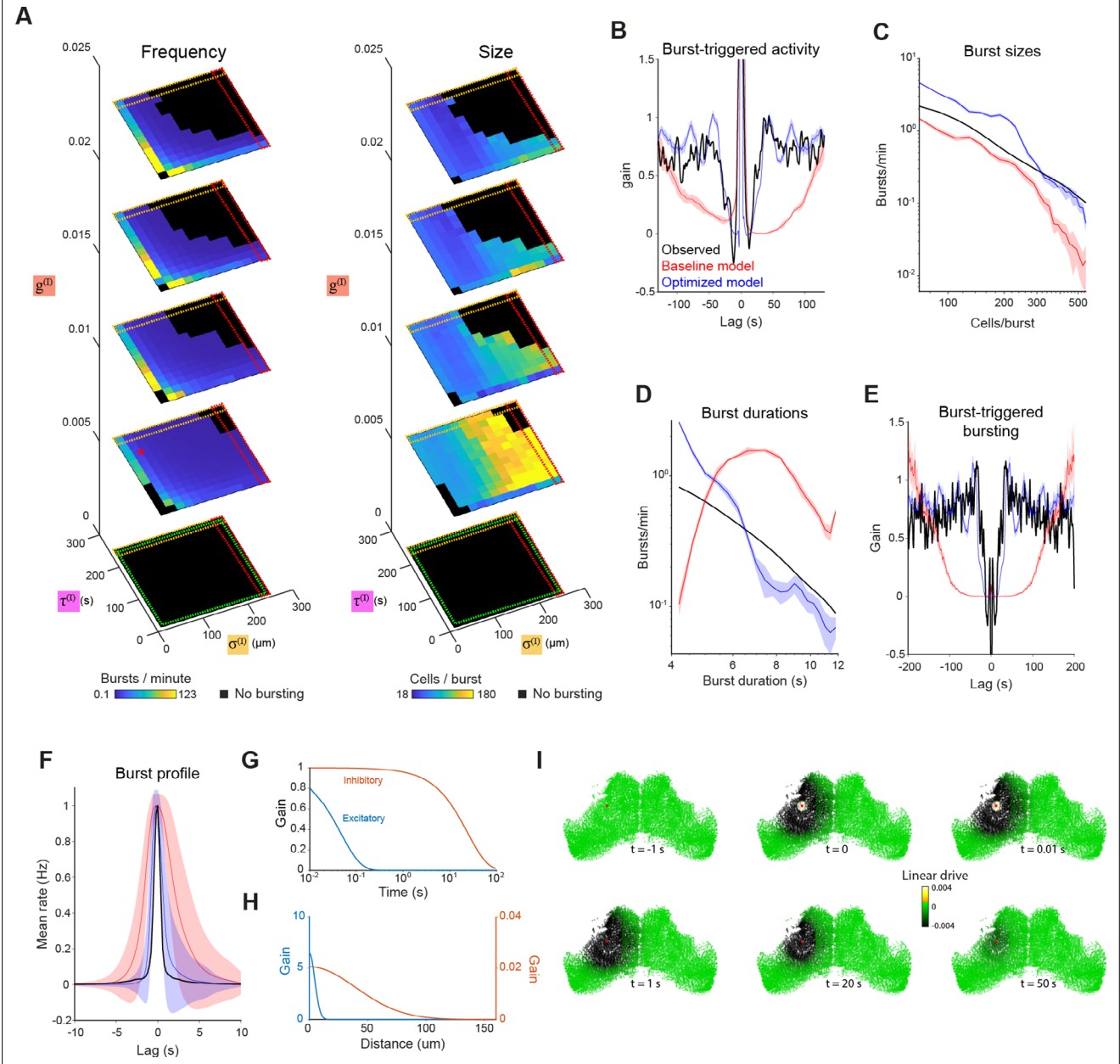

**Figure 3.** Exploration and optimization of model parameters. (**A**) Burst rate (left) and number of participating neurons (right) as a function of inhibitory space constant ($\sigma^{(I)}$), inhibitory time constant ($\tau^{(I)}$) and inhibitory gain ($g^{(I)}$). Red star indicates the parameters of the baseline model (as per *Figure 2*), and black regions indicate parameter combinations that failed to produce bursting. Dashed squares indicate regions approaching uniform inhibition (red), non-decaying inhibition (orange) and lack of inhibition (green). (**B–D**) Model simulation evaluated against optimization objectives: post-burst activity suppression (B), burst sizes distribution (C) and burst durations distribution (D). (**E–F**) Model simulation vs experimental data for two additional features: Burst-triggered burst participation (E, see *Figure 1L*) and burst temporal profile (F, see *Figure 1H*). (**G–H**) Temporal (G) and spatial (H) profiles of the intercellular interactions in the optimized model. (**I**) Timeseries illustrating the influence of a single spike (in cell marked with red spot) on the linear drive of surrounding cells in the optimized model network. Plots show mean and shaded areas indicate SD for n=10 simulation runs.

The online version of this article includes the following source data and figure supplement(s) for figure 3:

**Source data 1.** Data provided as a MATLAB structure.

**Figure supplement 1.** Model variations and optimization using EMOO.

interaction to balance recurrent excitation. Very large values of $\sigma^{(I)}$, approach the condition where inhibition is independent of distance: This resulted in very infrequent but large bursts (*Figure 3A*, red dashed line), as the entire network simultaneously became non-excitable and then synchronously recovered. Increasing the temporal decay constant ($\tau^{(I)}$) produced rare, small bursts (*Figure 3A*, orange dashed line) by prolonging the recovery time. We also confirmed the capacity of the model to produce bursting with an exponential rather than Gaussian relationship between distance and connection strength (*Figure 3—figure supplement 1A*), and the robustness of the bursting phenomenon to stochastic non-uniformities in connections by pruning random subsets of excitatory and inhibitory connections (*Figure 3—figure supplement 1B–C*). These analyses indicate that bursting is a robust phenomenon but that bursting statistics are modulated by the spatiotemporal parameters defining recurrent interactions.

Next, we sought to optimise the model parameters such that the stochastic, emergent property of bursting would best match tectal bursting statistics. We started with the set of parameters used in the previous section ('baseline model'), which generated bursting but deviated from biologically observed statistics with regard to frequency, size and duration of burst events (*Figure 2C-D*). We then used evolutionary multi-objective optimization (EMOO, *Kalyanmoy, 2001*, *Figure 3—figure supplement 1D* and Materials and methods) to tune the parameters according to three target objectives, namely the distribution of burst sizes (*Figure 1I*) and durations (*Figure 1J*), and the timecourse of burst-triggered activity suppression (*Figure 1K*). With each successive generation, EMOO produces a population of models that better approximates the Pareto optimal set of solutions (Pareto front), that is the set of models where no individual model performs better than any other in all three objectives (*Figure 3—figure supplement 1E*). Following thirty EMOO generations, we chose the solution at the elbow of the estimated Pareto front (the model closest to the origin in the z-scored loss space, *Figure 3—figure supplement 1E*, green), and used it as a starting point for a local search to fine-tune the parameters.

The resulting optimized model was successful in reproducing the bursting statistics of the zebrafish tectum (*Figure 3—figure supplement 1E*, blue). This was true for both the features used as optimization objectives (*Figure 3B–D*), as well as two that were not used for parameter tuning, namely burst-triggered burst participation probability (*Figure 3E*) and the burst temporal profile (*Figure 3F*).

The parameter values of the optimized model support the idea that bursting in OT arises from distinct spatiotemporal patterns of excitatory versus suppressive network interactions. Excitatory interactions are strong, short range ($\sigma^{(E)} = 4.5\mu m$) and decay rapidly ($\tau^{(E)} = 0.05s$), whereas suppressive interactions extend over longer distances ($\sigma^{(I)} = 40\mu m$) and decay slowly ($\tau^{(I)} = 24.1s$) and thus integrate activity for long periods of time (*Figure 3G–H*). As a result, a single spike in one neuron strongly excites its immediate neighbours for a brief moment (<1 s), while slightly reducing the excitability of a large part of the tectal hemisphere for tens of seconds (*Figure 3I*).

Because our optimized network model could reproduce spontaneous activity with biologically realistic statistics, we reasoned that the recurrent interactions it encapsulates may provide useful insight into the relationship between recent neuronal activity and the internal state (excitability) of the tectal network. Therefore, we next explored the extent to which model-inferred network state could account for variability in stimulus-evoked activity.

## Model-inferred network state accounts for variability in visually evoked tectal activity

Fluctuations in the state of excitability of neuronal populations caused by recent network activity is likely to contribute to variability in neural representations of sensory stimuli. To explore this, we applied our network model to recorded tectal activity data in order to estimate the incidental state of the biological network immediately prior to individual visual stimulus presentations and thereby predict the influence of ongoing activity upon stimulus-evoked responses.

We presented zebrafish with small moving spots, which evoke robust tectal activity (*Figure 4A*, *Niell and Smith, 2005*; *Bianco and Engert, 2015*). Spots swept horizontally across the visual field at two elevations, one slightly below and one above the horizon and to mitigate interactions between consecutive stimuli we initially used a long inter-stimulus interval (ISI, 5 min). The responses of tectal neurons displayed retinotopic mapping and direction selectivity (*Figure 4B*, *Stuermer, 1988*; *Niell and Smith, 2005*; *Hunter et al., 2013*).

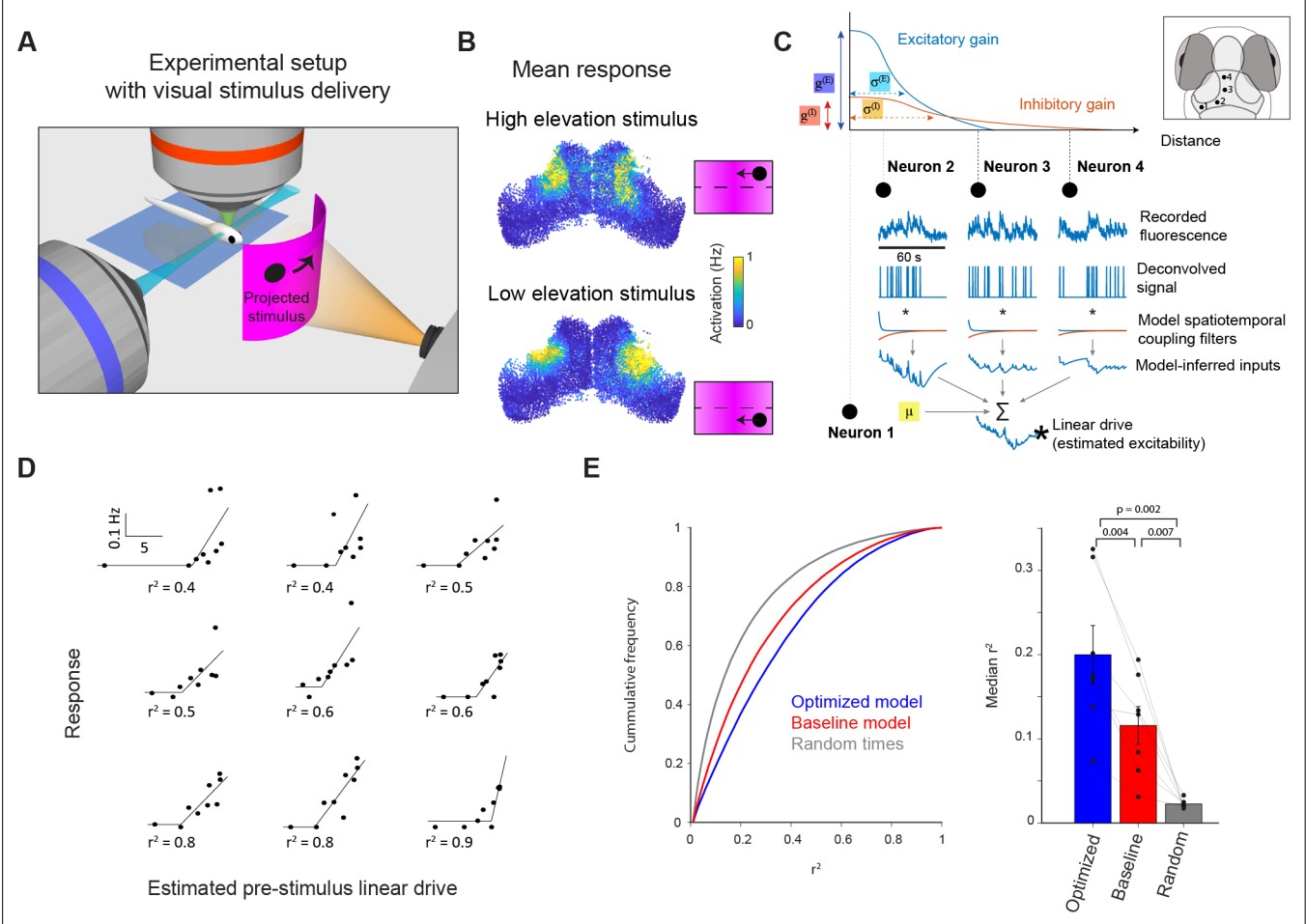

**Figure 4.** Incidental network state influences visually evoked responses. (**A**) Visual stimuli comprising prey-like moving spots were presented to larvae during light-sheet imaging. (**B**) Average firing rate of tectal cells during presentation of leftwards moving spots at high (top) and low (bottom) elevation. (**C**) Estimation of neuronal excitability (linear drive) based on recent ongoing activity and model parameters. (**D**) Examples of nine neurons from one fish for which model-estimated pre-stimulus linear drive explained visually evoked activity. Each point corresponds to one stimulus presentation. $r^2$ values shown for a threshold-linear fit. (**E**) Left: Cumulative distribution of $r^2$ values for threshold-linear fits of visual responses as a function of model-estimated linear drive (with a positive slope constraint). N=56,000 neurons in 7 fish. Linear drive was estimated using the optimized or baseline model parameters (blue and red, respectively) or from randomly chosen sequences of ongoing activity, using the optimized model parameters (grey). Right: Median $r^2$ values across neurons from each animal. Only cells with non-zero median visual response were included. Thin lines indicate median values for individual fish, bars show mean ± SEM across fish, p-values for paired t-test.

The online version of this article includes the following source data and figure supplement(s) for figure 4:

**Source data 1.** Data provided as a MATLAB structure.

**Figure supplement 1.** Contribution of model parameters to response prediction.

Using this data, we examined the relationship between response variability and model-estimated network state. First, we used our model to estimate the incidental state of OT neurons immediately prior to each stimulus presentation, based on the recent history of ongoing activity in the network. To do this, spikes inferred from calcium imaging data during the minute prior to stimulus onset (specifically, –60 to –5 s before stimulus onset) were spatiotemporally filtered and summed for each cell using the model parameters (*Figure 4C*, Materials and methods). In this way, we estimated the `linear drive` of biological neurons (treated here as equivalent to their excitability), in the same way as for the previous simulations, except that we used recorded, rather than simulated, activity.

Model-predicted excitability explained a substantial fraction of the trial-to-trial variability of visually evoked neural activity. Specifically, we found that for individual neurons, a threshold-linear function described the visual responses of the cell across single trials in terms of the pre-stimulus

linear drive, estimated using the model, for those same trials (*Figure 4D–E*). Model-inferred linear drive explained response variability significantly better than a 'null' estimate of network state, using randomly sampled timepoints, or states estimated using the baseline model parameters (*Figure 4E*, Materials and methods).

These results suggest that the recurrent interactions implemented within our network model can predict incidental network state and thereby account for a fraction of the trial-to-trial variability of visually evoked tectal activity. Because the excitatory interaction is short-lived, this effect of recent ongoing activity is largely determined by spatiotemporal properties of the more prolonged inhibitory interaction. For example, the improved performance between the baseline and optimized model was largely due to the decrease in $\tau^{(I)}$ (from 180 to 24 s), with a small contribution from the increase in $\sigma^{(I)}$ (from 20 μm to 40 μm, *Figure 4—figure supplement 1*). In sum, recent network activity in the vicinity of a neuron tends to supress its excitability, resulting in a relatively reduced response to visual sensory input.

## Ongoing activity is supressed following visually evoked activity

Next, we explored the reciprocal interaction, namely if sensory stimuli exert a lasting effect on the state of the network that consequently perturbs ongoing activity.

First, we used our model to predict the effects of visual input on network state. We expanded the model to incorporate external (sensory) input such that model neurons were activated according to the visual receptive fields of their corresponding tectal neurons. We estimated receptive fields by fitting Gaussian functions to the measured activity of each cell as a function of stimulus angle (*Figure 5A*, Materials and methods). Receptive fields collectively covered visual space, with an average width (2.5σ) of 38°±24° (mean ± SD, *Figure 5B*), comparable to previous reports (*Niell and Smith, 2005*). Because visual space is retinotopically mapped within OT, neurons responsive to high and low elevation stimuli occupied distinct anatomical locations (*Figure 5C*, Materials and methods).

Running the network simulation with sensory input predicted a substantial, long-term reduction in tectal activity for tens of seconds following visual stimulation (*Figure 5D*). This prediction is concordant with the long inhibition time constant in the model (24.1 s) and, as a result of the spatially weighted recurrent connectivity, was biased to recently stimulated regions of OT. Thus, cells responsive to the recently presented stimulus were predicted to show a greater suppression of ongoing activity in the period following stimulus offset as compared to neurons responsive to the other elevation (*Figure 5D*). This spatial-selective suppression of activity was explained by a lasting state of low linear drive extending beyond the duration of the sensory stimulus (*Figure 5F*, *Video 3*).

To test this prediction, we repeated the same analysis with calcium imaging data from five fish. As predicted by the model simulation, neurons responsive to the recently presented stimulus showed a marked reduction of ongoing activity for tens of seconds, whereas neurons responsive to the other elevation showed a lesser suppression of post-stimulus activity (*Figure 5E*).

In sum, our network model can account for bidirectional interactions between neuronal activity and network state that lead to interactions between ongoing and stimulus-evoked tectal activity. Ongoing activity shapes the incidental state of neuronal excitability, which in turn modulates visual responses, while these responses exert a long-lasting suppressive effect upon network excitability and ongoing activity.

## Long-lasting activity-dependent suppression explains adaptation of visual responses

These observations lead to the prediction that visual stimuli will have a lasting effect on the state of the tectal network that may influence subsequent sensory responses. Thus, we next used network modeling and calcium imaging to explore how the recent history of sensory stimulation impacts visually evoked activity and behavior.

Based on the timescales and spatial ranges apparent in tectal dynamics, we designed an experimental paradigm incorporating a mixture of temporal and spatial relationships between visual stimuli. During 30 min 'stimulation blocks', small moving spots were presented at two different elevations (out of a possible three). One stimulus was presented at high frequency ('common', 30 s inter-stimulus interval, ISI) and the second at low frequency ('deviant', 5 min ISI). Each deviant stimulus was followed

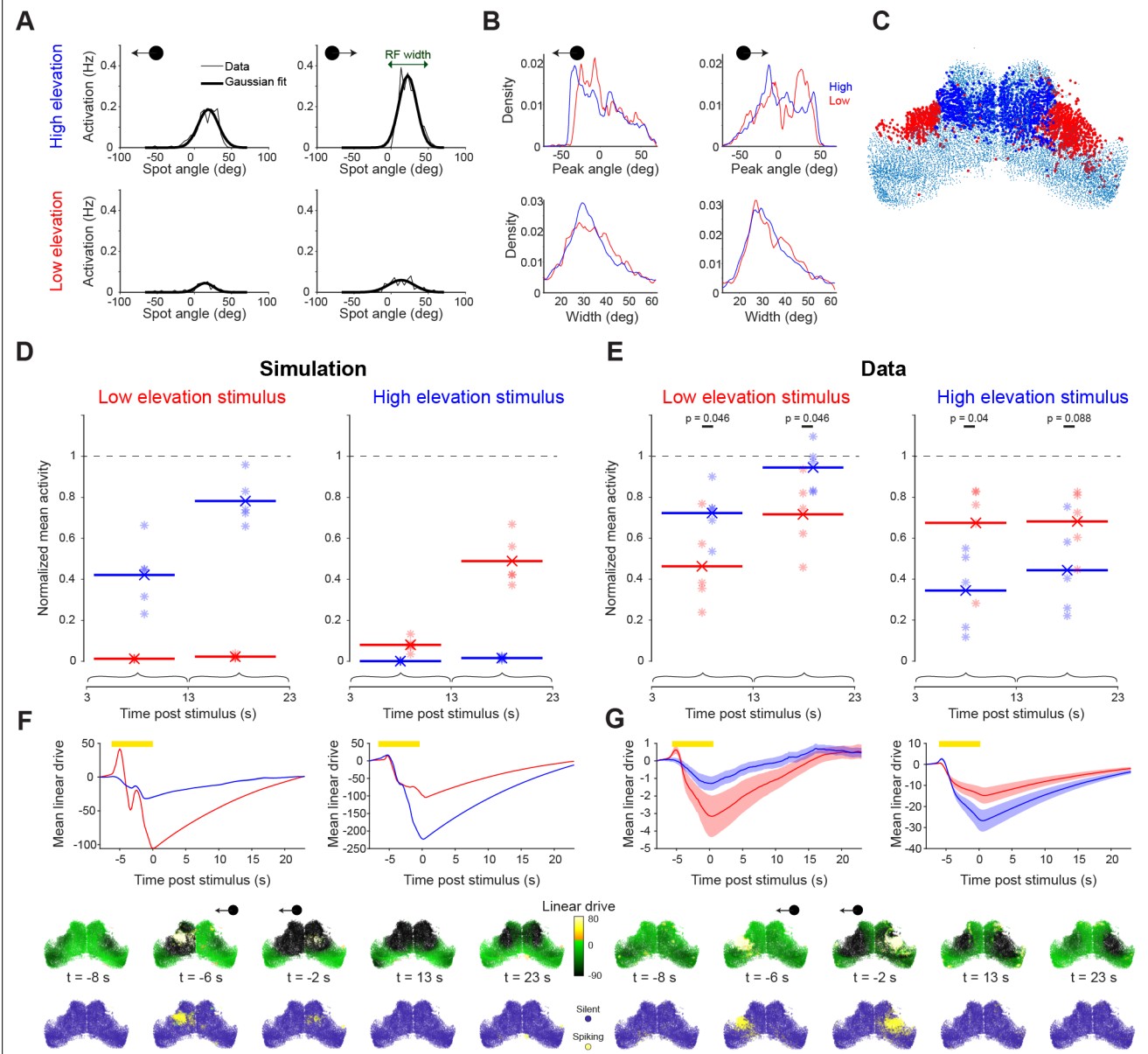

**Figure 5.** Visual stimulation is followed by long lasting, spatially selective suppression of tectal activity. (**A**) Visually evoked activity of an example neuron showing mean spike counts (thin line) and Gaussian fit (thick line) for moving spots at high (top) and low (bottom) elevation and moving leftward (left) or rightward (right). This neuron shows modest direction selectivity. (**B**) Distribution of receptive field centres (top) and widths (bottom) for cells responsive to moving spots at high (blue) and low (red) elevation and moving leftward (left) or rightward (right), with good Gaussian fits ($r^2 > 0.8$). (**C**) Locations of cells responsive to high (blue) and low (red) elevation stimuli. (**D–E**) Post-stimulus activity for simulated (D) and recorded (E) neurons. Activity is shown for cells responsive to high (blue) and low (red) elevation and integrated across two time-windows following the cessation of the visual stimulus. Each point indicates a single simulation run or experimental session, lines indicate average values. Values are normalized by the mean ongoing activity. (**F–G**) Top: Mean simulated (F) and estimated (G) linear drive of cells responsive to high (blue) or low (red) elevation during and after the presentation of a low (left) or high (right) elevation stimulus (yellow bar). Bottom: Simulated (F) and estimated (G) linear drive and spiking for single example trials at the indicated times from stimulus offset.

The online version of this article includes the following source data for figure 5:

**Source data 1.** Data provided as a MATLAB structure.

by a 90 s break. Stimulation blocks were separated by 30 min 'rest blocks', during which no stimuli were presented (*Figure 6A*).

We first evaluated the model prediction for this experimental protocol by simulating it five times. To take into account adaptation of retinal inputs, we imaged the tectal neuropil in *isl2b:GAL4;UAS:SyGC6s*

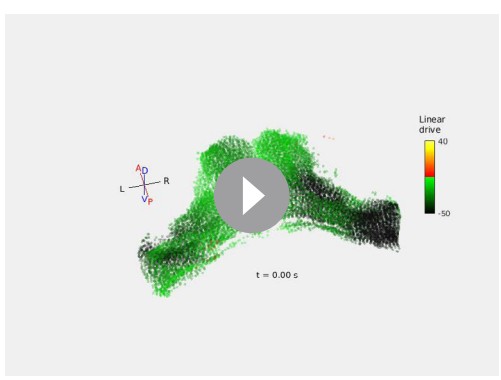

**Video 3.** Simulated linear drive during and after visual stimulus. Time-course of linear drive to model cells in a simulation of an evoked response to a moving spot stimulus (black circle).

https://elifesciences.org/articles/78381/figures#video3

fish expressing GCaMP6s at the synaptic terminals of retinal ganglion cells (RGCs), and modulated the input gain for the simulated high-frequency stimulus accordingly (*Figure 6—figure supplement 1* and Materials and methods). The model predicted an adaptation of tectal visual responses to the common stimulus, but not to the deviant stimulus (*Figure 6A*). For example, during the 2nd stimulation block, high and low elevation stimuli were presented as common and deviant stimuli, respectively (*Figure 6A*, blue and red). Adaptation reduced the mean response of model cells tuned to the high elevation stimulus to about 70% of their initial amplitude followed by a partial recovery during the 90 s breaks (*Figure 6A*, stars). This is due to a combination of gradual input (RGC) adaptation and additional activity-dependent suppression intrinsic to the tectum (*Figure 6—figure supplement 1C–D*). Conversely, the response to the low elevation stimulus remained approximately constant.

To test these predictions against experimental observations, we used the same stimulation protocol with five fish (*Figure 6B*). In agreement with the model prediction, high-frequency stimulus presentation caused a spatially specific response adaptation to the common stimulus. The adaptation pattern matches the predicted combined effect of RGC adaptation and intrinsic tectal dynamics (*Figure 6A*), rather than either one alone (*Figure 6—figure supplement 1C–D*).

To examine how this response modulation varies across cells, we compared the average response of individual cells during the 2nd stimulation block to their baseline response (specifically, their average response to low-frequency stimulation, large circles in *Figure 6B*). The response to the common (high elevation) stimulus was attenuated in most cells (*Figure 6C–D*, cyan), and attenuation magnitude was linearly correlated with the amplitude of the baseline response (*Figure 6E*). By contrast, response modulation to the deviant (low elevation) stimulus was less consistent (*Figure 6C–D*, yellow), although some attenuation was observed in cells that were strongly activated by the common stimulus (*Figure 6E*). These results show that the common stimulus drives a subset of tectal cells to a supressed state, which modulates their subsequent responses to either stimulus.

To conclude, OT displays experience-dependent response modulation as predicted by our network model. This effect allows for integration of past stimuli and suppression of subsequent responses to similar stimuli.

## Activity-dependent changes in tectal state underlie variability in prey-catching behavior

Lastly, we examined how activity-dependent changes in network state relate to visually guided prey-catching behavior, which is mediated by OT (*Gahtan et al., 2005*; *Muto et al., 2013*; *Bianco and Engert, 2015*; *Helmbrecht et al., 2018*). Because hunting routines of larval zebrafish invariably commence with eye convergence, we used the occurrence of convergent saccades (CS) to identify prey-catching behavioral responses (*Figure 7A* inset, *Bianco et al., 2011*; *Trivedi and Bollmann, 2013*; *Bianco and Engert, 2015*).

We first analyzed 'spontaneous' CSs, which occasionally occur in the absence of visual stimuli (*Bianco and Engert, 2015*), to assess the link between ongoing tectal activity and behavioral output. By examining activity in 1 s time windows centred on CSs ('CS-associated activity', *Figure 7A*, orange shaded area), we found that spontaneous CSs are associated with an increase in tectal activity, particularly in the anterior tectum (*Figure 7A–B*, anterior tectum p=0.002, n=11 fish, Wilcoxon signed rank test, anterior vs posterior activity p=0.02, n=11 fish, Wilcoxon signed rank test). Moreover, CS-associated activity was lateralized in accordance with the direction of behavior. We showed this by comparing activity between the tectal hemispheres for spontaneous CSs that directed gaze to the

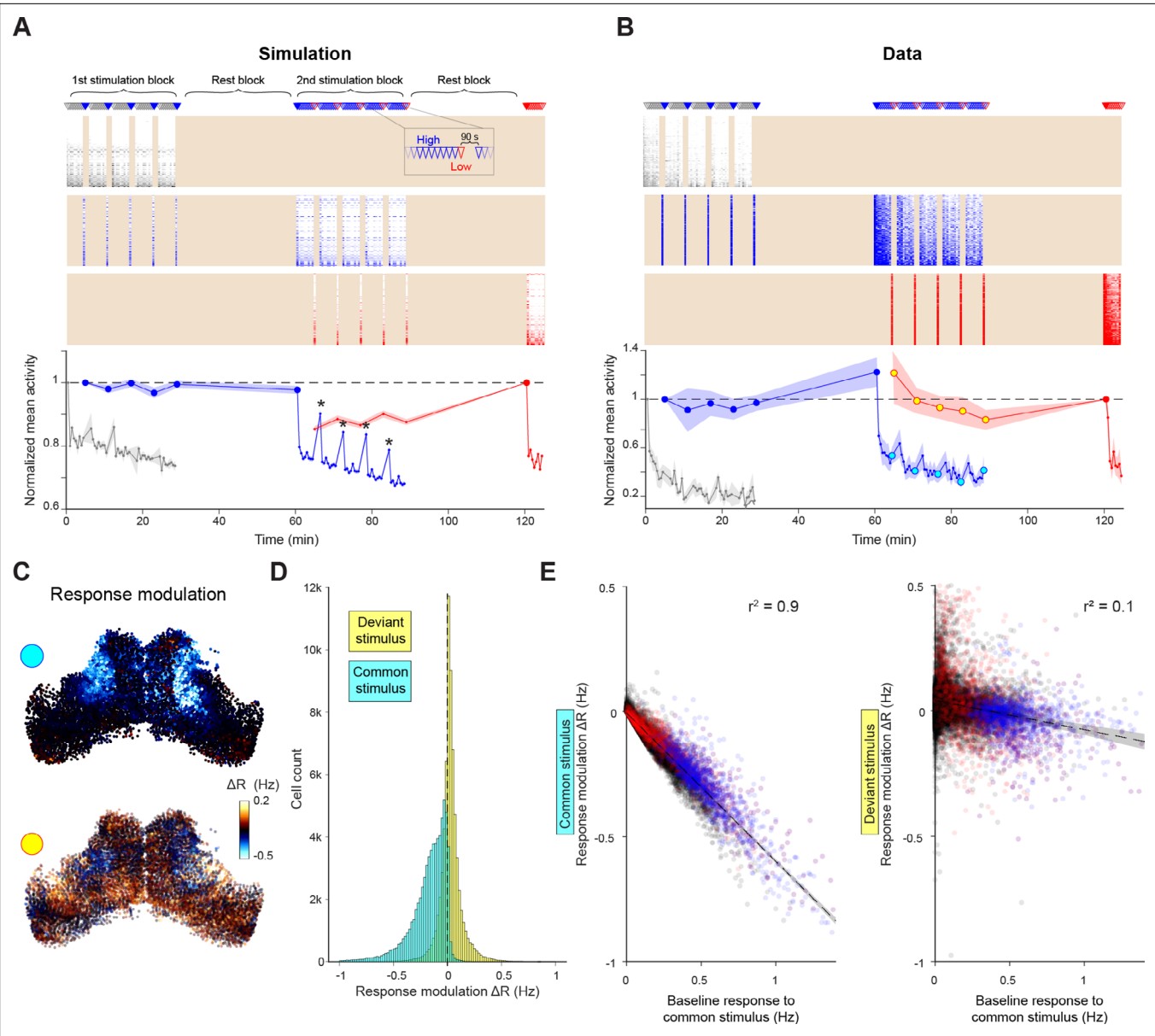

**Figure 6.** Selective adaptation to visual stimuli. (**A–B**) Simulated (A) and recorded (B) responses of cells tuned to very low (grey), high (blue) and low (red) elevation stimuli. Top: Stimulation protocol. Middle: Raster of responses of individual cells (activity in inter-stimulus periods is excluded, shading). Bottom: Mean response across cells tuned to each elevation, normalised by first response. Large symbols indicate epochs used to establish the `baseline` response for analysis in (C-E), stars indicate partial response recovery following 90 s breaks. Shaded areas indicate SEM for n=5 fish or simulation runs. (**C**) Top: Response modulation for the high elevation stimulus. For every OT cell, we compare the baseline response (large blue circles in (B)) to the 2nd stimulation block (cyan circles in B) where the stimulus is presented as a common stimulus. Bottom: Response modulation for the low elevation stimulus, comparing baseline responses (large red circle in (B)) to the 2nd stimulation block (yellow circles in (B)) where it is presented as a deviant stimulus. (**D**) Distribution of single-cell response modulation for the common (cyan) and deviant stimulus (yellow), (n=67,750 cells from 5 fish). (**E**) Single cell response modulation for common (left) and deviant stimulus (right) as a function of the baseline response to the common stimulus. Blue and red spots indicate cells tuned to the common and deviant stimuli, respectively. Shaded areas indicate linear regression confidence intervals for $\alpha=10^{-10}$. See also *Figure 6—figure supplement 1*.

The online version of this article includes the following source data and figure supplement(s) for figure 6:

**Source data 1.** Data provided as a MATLAB structure.

**Figure supplement 1.** Imaging of retinal ganglion cell (RGC) axonal arbors and additional network simulations excluding retinal adaptation.

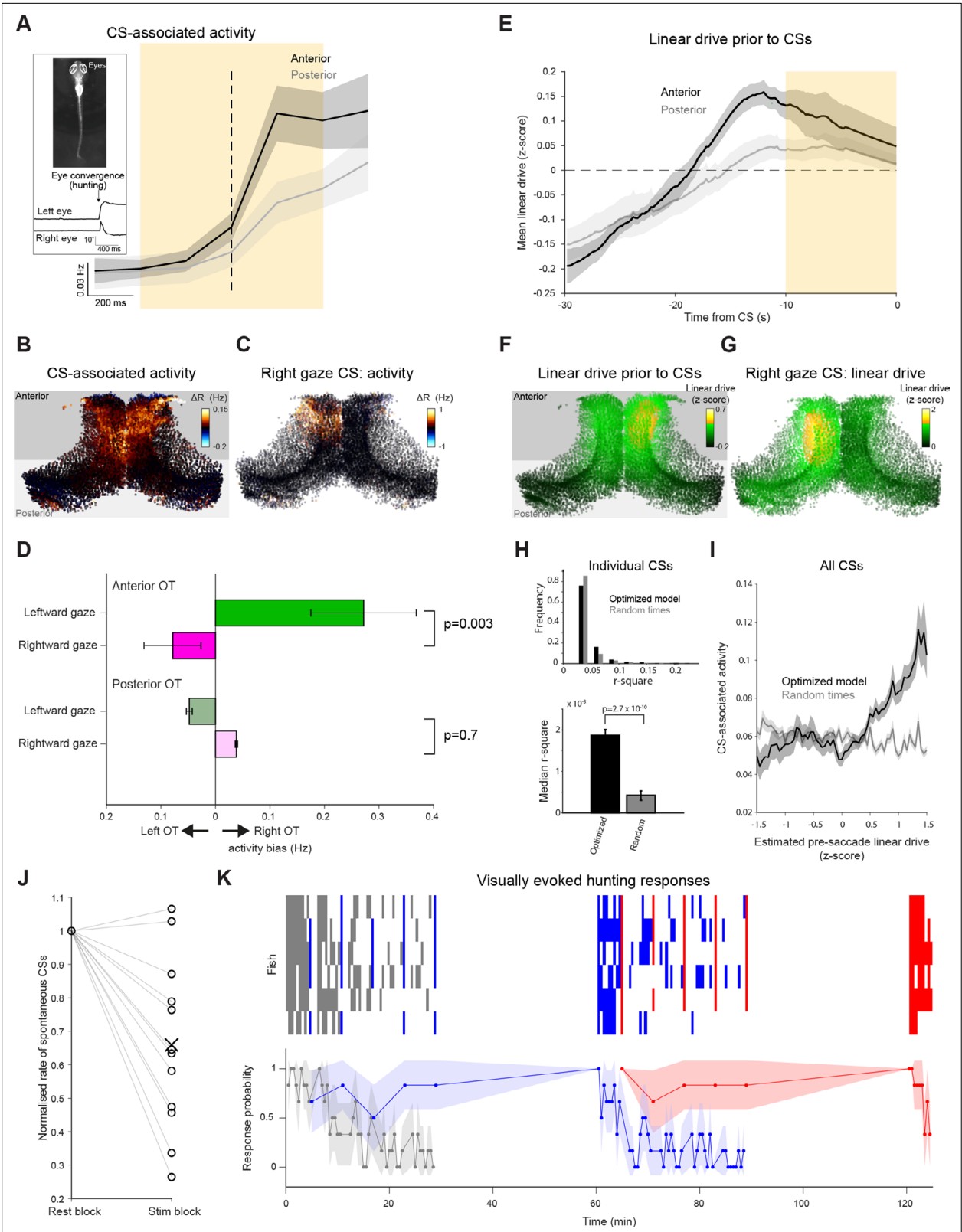

**Figure 7.** Prey-catching behavior is modulated by tectal network state and stimulus history. (**A**) Activity of neurons in the anterior (black) and posterior (grey) OT, triggered on spontaneous CSs. Mean with shaded areas indicating SEM (n=11 fish), dashed line indicates CS time. Inset: Eye tracking and an example of a detected convergent saccade. (**B**) Mean baseline-subtracted activity of OT cells during 1 s windows centred on spontaneous CSs (orange rectangle in (A), n=66 CSs in a single fish). (**C**) Baseline-subtracted activity for a single spontaneous CS with rightward post-saccade gaze angle.

*Figure 7 continued on next page*

*Figure 7 continued*

(**D**) Comparisons of activity between OT hemispheres for lateralized CSs with leftward (green) and rightward (magenta) post-saccade gaze angle. Error bars indicate SEM (n=408 CSs from 11 fish). (**E**) Linear drive in the anterior (black) and posterior (grey) OT, prior to CSs, estimated using optimized model parameters and 60 seconds of pre-saccade ongoing activity. Mean with shaded area indicating SEM (n=11 fish). Dashed line indicates baseline linear drive. (**F**) Linear drive of OT cells during the 10 s prior to spontaneous CSs (orange rectangle in (E), in the same fish depicted in (B), n=66 CSs). (**G**) Linear drive prior to the spontaneous CS shown in (C). (**H**) Top: Distribution of r$^2$ values for linear fits of CS-associated activity as a function of estimated linear drive (with a positive slope constraint). Linear drive ('excitability') of individual neurons was estimated using optimized model parameters and 60 s of pre-saccade ongoing activity (black) or randomly chosen sequences of ongoing activity (grey). 680 CSs from 11 fish. Bottom: median r$^2$ values, error bars indicate ±1%, p-values for a Wilcoxon signed rank test. (**I**) For each of the CSs, the mean across neurons of CS-associated activity was computed for bins of estimated linear drive. Shaded area indicates bin-wise SEM. (**J**) Rate of spontaneous CSs during rest blocks (more than 120 s since the last visual stimulus presentation) vs visual stimulation blocks, normalised by rest block rate. Each line indicates one fish. (**K**) Prey-catching responses evoked by very low (grey), high (blue), and low (red) elevation stimuli, presented according to the protocol described in *Figure 6*. Top raster show responses from each of six animals (rows) and lower trace shows mean response probability with 90% confidence interval.

The online version of this article includes the following source data for figure 7:

**Source data 1.** Data provided as a MATLAB structure.

left or right (>10° post-saccade gaze angle). In anterior (but not posterior) OT, CS-associated activity was significantly biased towards the hemisphere contralateral to post-saccadic gaze (*Figure 7C–D*, p=0.003 and 0.7 for anterior and posterior OT, n=408 CSs, two-way ANOVA). These findings are in agreement with previous reports linking anterior OT activity with prey-catching behavior directed to the contralateral side (*Fajardo et al., 2013*; *Bianco and Engert, 2015*).

These observations are compatible with intrinsic tectal dynamics contributing to the generation of spontaneous prey-catching responses. Alternatively, CS-associated activity could be a consequence of the eye movement (visual or proprioceptive sensory feedback), or represent a motor efference copy. Therefore, to investigate if tectal activity is likely to cause spontaneous behavior, we used our network model to estimate the state of tectal neurons (their linear drive) immediately prior to spontaneous CSs based on the recent activity history of the network (same method as above, *Figure 4*). This analysis showed that linear drive in anterior OT was significantly elevated prior to CSs (*Figure 7E–F*, pre-CS vs baseline p=0.004, n=11 fish, one-tailed t-test), and this difference was larger compared to posterior OT (p=0.04, n=11 fish, one-tailed paired t-test). At a single-cell level, pre-CS linear drive was positively correlated with CS-associated activity, both for individual CSs (*Figure 7H*) as well as for data pooled over all spontaneous CSs (*Figure 7I*). These results are compatible with ongoing activity in the tectal network modulating the excitability state of OT neurons and contributing to the generation of spontaneous prey-catching behavior.

Because we have shown that the recent history of visual stimuli has lasting effects upon tectal activity, we next asked how past experience of visual inputs modulates spontaneous and visually evoked prey-catching behavior. To do this, we used the mixed-ISI stimulation protocol (*Figure 6*), while tracking behavior.

When we analysed spontaneous behavior, we found that the rate of spontaneous CSs in the 2 min that followed any visual stimulus was reduced by approximately one-third compared to rest blocks (*Figure 7J*, n=13 fish, p=0.002, Wilcoxon signed rank test). This suppression of behavior is concordant with the lasting reduction in excitability state and ongoing tectal activity described above. Moreover, the probability of visually evoked prey-catching was modulated by stimulus history in a qualitatively similar way to visually evoked tectal activity. Specifically, response rates declined for common, but not deviant stimuli (*Figure 7K*, compare *Figure 6B*).

Taken together, these observations suggest that the state of the tectal network, reflecting the integration of previous sensory inputs as well as ongoing activity, impacts both spontaneous and visually evoked prey-catching behavior.

## Discussion

In this study, we combined functional imaging and computational modeling to investigate how variability in sensory encoding and visuomotor behavior might arise from activity-dependent fluctuations in network state. Based on observations of ongoing, 'spontaneous', activity, we inferred a recurrent circuit motif incorporating fast, local excitation and long-lasting, activity-dependent suppression,

which was sufficient to reproduce the occurrence and statistics of tectal bursting. Applying these same recurrent interactions to recent activity inferred from calcium recordings gave us a means to predict the (usually unseen) excitability state of tectal neurons and thereby explain a portion of the variability in visually evoked responses. Prolonged recurrent suppression explained the lasting effect of sensory stimuli on ongoing tectal dynamics as well as spatially selective visual response adaptation. Finally, model-predicted network state explained variability in prey-catching responses, suggest these recurrent interactions are relevant to tectally mediated behavior. In sum, our findings suggest that a recurrent interaction motif can account for bidirectional interactions between neuronal activity and network state, which contribute to variability in sensory encoding and behavior.

## Modeling approach

Motivated by the idea that ongoing activity is both a cause and consequence of changes in the excitability state of neural populations, we inferred a minimal recurrent circuit motif that was sufficient to generate stochastic activity sequences that closely resembled ongoing tectal activity. We achieved this by combining classical neural field models (*Amari, 1977*), where two opposing interactions are deterministic functions of intercellular distance, with the stochasticity of spiking Linear-Nonlinear-Poisson (LNP)-type models (*Pillow et al., 2005*; *Pillow et al., 2008*). Despite being defined by only seven parameters, this minimal motif supported the emergence of stochastic tectal bursting with biologically realistic statistics. The model also explained some of the variability in visually evoked activity, the effect of that activity on network state, visual response adaptation and spatially specific behavioral habituation, suggesting this recurrent interaction motif is a dominant feature of the optic tectum.

However, applying the same connectivity rule, with identical parameters, across every tectal neuron is certainly an oversimplification and our model should not be regarded as a detailed approximation of tectal connectivity. Indeed, although it qualitatively reproduced several biological observations, model predictions deviated quantitatively and more expressive models that accommodate cell-specific connectivity parameters are expected to provide more accurate quantitative predictions. Future extensions to the model, incorporating known heterogeneities in neurotransmission, morphology and intrinsic properties and with many more free parameters, should expand the accuracy and breadth of the model assuming such additions are adequately constrained by functional and/or anatomical data. In this regard, recent studies in zebrafish have characterised various aspects of molecular-genetic, morphological, and physiological diversity in OT (*Helmbrecht et al., 2018*; *Antinucci et al., 2019*; *Shainer et al., 2022*) and high-quality ultrastructural data is now available, allowing definitive reconstruction of synaptic connectivity (*Svara et al., 2022*). The model can also be extended to incorporate interconnections between OT and many other brain regions (e.g. nucleus isthmi, *Henriques et al., 2019*), guided by data from whole-brain activity recordings (*van der Plas et al., 2023*) and brain wide anatomical atlases (*Marquart et al., 2015*; *Randlett et al., 2015*; *Hildebrand et al., 2017*; *Shainer et al., 2022*).

## Bursting activity and tectal connectivity

Recent studies have suggested that bursting activity observed in OT represents recruitment of subnetworks with enhanced connectivity, or neuronal 'assemblies' (*Romano et al., 2015*; *Avitan et al., 2017*; *Marachlian et al., 2018*; *Mölter et al., 2018*; *Diana et al., 2019*). At the core of this model is the idea that to explain OT bursting there must be nonuniformities in connectivity in which cells forming an assembly are preferentially interconnected and operate as a local attractor network. However, here we show that biologically realistic localised bursting *can* emerge in a network in which, by design, there are no assemblies; rather the same connectivity motif (in which recurrent interactions smoothly decay with distance) is implemented identically for every cell. Although, as discussed above, our model is certainly and oversimplification and does not rule out the existence of assemblies, it highlights that localised bursting, by itself, is insufficient evidence for such heterogeneities in connectivity.

## The biological implementation of recurrent interactions

The recurrent interaction motif in our network model predicts that the activity of a given OT neuron strongly but transiently increases the excitability of its immediate neighbours, while causing sustained suppression of a broader population of neurons in its vicinity (*Figure 3I*). What biological mechanisms might underlie such interactions?

The short time scale we estimated for excitatory interactions ($\tau^{(E)} = 0.05s$) is compatible with direct, recurrent synaptic connectivity in OT. Such connections have been demonstrated by intracellular recording of delayed excitatory post-synaptic potentials in tadpole tectum following optic tract stimulation (*Pratt et al., 2008*). Delayed glutamatergic potentials were similarly recorded in slices from rat superior colliculus and are thought to mediate pre-saccadic bursting (*Saito and Isa, 2003*).

By contrast, the inhibitory interactions in our model caused activity-dependent suppression lasting tens of seconds ($\tau^{(I)} = 24.1s$), too long to be accounted for by direct ionotropic synaptic inhibition. This interaction might be mediated by pre-synaptic depression, which can persist for tens of seconds in the avian OT (*Luksch et al., 2004*). Alternatively, slow suppression might be mediated by activity-dependent changes in intracellular properties such as ionic concentrations or membrane conductances, potentially lasting for several minutes (*Desai et al., 1999*; *Karmarkar and Buonomano, 2006*; *Gasselin et al., 2015*; *Zylbertal et al., 2017b*).

Long-term suppression might also derive from activity in inhibitory populations external to OT. There is currently no evidence to suggest that tectal interneurons are involved in such suppression, since their visually evoked activity appears to be limited to transient responses (*Preuss et al., 2014*; *Dunn et al., 2016*). However, brain regions interconnected with OT, such as GABAergic populations in the vicinity of the isthmic hindbrain (*Gebhardt et al., 2019*; *Henriques et al., 2019*), might provide long lasting feedback inhibition.

## Tectal dynamics and visuomotor transformation underlying hunting

Our results indicate that the state of the tectal network is predictive of prey-catching response probability, in agreement with the role of OT in generating premotor signals for this visuomotor behavior (*Herrero et al., 1998*; *Gahtan et al., 2005*; *Bianco and Engert, 2015*; *Antinucci et al., 2019*). What adaptive purpose might the network interactions we describe play during natural behavior?

The optic tectum (and its mammalian homolog the superior colliculus), is thought to be involved in detection and selection of salient inputs and generation of rapid behavioral responses (*Boehnke and Munoz, 2008*; *Dutta and Gutfreund, 2014*). The experience-dependent activity suppression we observe is compatible with this role, as it filters out expected inputs which are less likely to be associated with immediate danger or prey-catching opportunity. Similarly, zebrafish OT exhibits response adaptation to looming stimuli that is thought to play a role in the habituation of escape behavior (*Marquez-Legorreta et al., 2019*). Comparable stimulus-selective adaptation and behavioral habituation has been described for auditory stimuli in barn owl OT (*Reches and Gutfreund, 2008*; *Netser et al., 2011*), and for overhead looming stimuli in the mouse superior colliculus (*Lee et al., 2020*). Notably, adaptation in the barn owl is selective for multiple auditory stimulus features such as interaural time and level differences, as well as frequency and amplitude. In the simple model we propose, the strength of interactions between neurons depends solely on Euclidean distance and therefore the spatial proximity of their receptive fields. An interesting avenue for future study will be to examine whether recurrent interactions weighted by similarity of feature tuning can account for such feature-selective adaptation.

## Materials and methods

### Animals

Zebrafish (*Danio rerio*) larvae were reared on a 14/10 hr light/dark cycle at 28.5 °C. For all experiments, we used zebrafish larvae homozygous for the *mitfa*[w2] skin-pigmentation mutation (*Lister et al., 1999*). For Ca2 +imaging experiments, we used larvae homozygous for *Tg(elavl3:H2B-GCaMP6s)*[jf5Tg] (*Vladimirov et al., 2014*, ZFIN ID: ZDB-ALT-141023–2). For imaging RGC axonal projections, larvae were double transgenic for *Tg(isl2b:Gal4)*[zc60Tg] (*Ben Fredj et al., 2010*, ZFIN ID: ZDB-ALT-101130–1) and *Tg(UAS:SyGCaMP6s)*[a155Tg] (*Dunn et al., 2016*, ZFIN ID: ZDB-ALT-160406–1). All larvae were fed *Paramecia* from 4 dpf onward. Animal handling and experimental procedures were approved by the UCL Animal Welfare Ethical Review Body and the UK Home Office under the

Animal (Scientific Procedures) Act 1986, under Home Office project licence 70/8162 awarded to Isaac Bianco.

## Light-sheet functional calcium imaging and behavioral tracking

For calcium imaging we used a custom-built digitally scanned light-sheet microscope. The excitation path included a 488 nm laser source (OBIS 488–50 LX, Coherent, Santa Clara, California), a pair of galvanometer scan mirrors (6210 H, Cambridge Technology, Bedford, Massachusetts) and objective (Plan 4 X, 4 x/0.1 NA, Olympus, Tokyo, Japan). A water-immersion detection objective (XLUMPLFLN, 20 x/1.0 NA, Olympus), a tube lens (*f*=200 mm), two relay lenses (*f*=100 mm) in a 4-f configuration, and sCMOS camera (Orca Flash 4.0, Hamamatsu, Hamamatsu, Japan) were used in the orthogonal detection path. For remote focusing (*Fahrbach et al., 2013*), an electrically tunable lens (ETL, EL-16–40-TC-VIS-20D, Optotune, Dietikon, Switzerland) was installed between the relay lenses, conjugate to the back focal plane of the objective. Volumes (375x410 x 75 µm) comprising 19 imaging planes spaced 4 µm apart, were acquired at 5 volumes/s. Each plane received laser excitation for 1ms (duty cycle 9%) resulting in average laser power at sample of 12.4 µW. To keep the observed population of neurons in each plane in focus throughout long imaging sessions, we implemented an automatic correction for slow drift in the Z direction. At the beginning of the experiment, we acquired two reference stacks, centred on two of the imaging planes, by incrementally biasing the Z scanning mirror and ETL in steps of 1.5% of their scan amplitude. During the course of the experiment, Z drift was estimated every 30 seconds by comparing recent images to these reference stacks, finding the reference images with the maximal XY cross-correlation, and averaging the two drift estimates. Every 5 min, the Z scan mirror and ETL signals were biased to offset any detected drift, according to the average of the ten most recent Z drift estimates.

For functional imaging, larval zebrafish were mounted in a custom 3D printed chamber (SLS Nylon 12, 3DPRINTUK, London, United Kingdom) in 3% low-melting point agarose (Sigma-Aldrich, St. Louis, Missouri) at 5 dpf and allowed to recover overnight before functional imaging at 6 dpf. Visual stimuli were back-projected (ML750ST, Optoma, New Taipei City, Taiwan) onto a curved screen forming the wall of the imaging chamber in front of the animal, at a viewing distance of ~10 mm. A colored filter (Follies Pink No. 344, Roscolux, Stamford, Connecticut) was placed in front of the projector to block green light from the collection optics. Visual stimuli were designed in Matlab (MathWorks, Natik, Massachusetts) using Psychophysics toolbox (*Brainard, 1997*). Stimuli comprised 10° dark spots on a bright magenta background, moving at 20°/s either left→right or right→left across ~110° of frontal visual space. Two or three elevation angles were used, calibrated for each fish during preliminary imaging by finding elevations separated by at least 15° that produce robust observable tectal activation (typically a very low elevation stimulus ~25° below the horizon, a low elevation stimulus ~10° below the horizon, and a high elevation stimulus ~5° above the horizon).

Eye movements were tracked during imaging experiments at 50 Hz under 850 nm illumination using a sub-stage GS3-U3-41C6NIR-C camera (Point Grey, Richmond, Canada). The angle of each eye was inferred online using a convolutional neural network (three 5x5 convolutional layers with 1, 1 and 4 channels, each followed by stride-2 max pooling layer, and a single fully connected layer), pre-trained by annotating images from multiple fish covering a wide range of eye positions. Eye movements were categorized as a convergent saccade if both eyes made nasally directed saccades within 150 ms of one another. Microscope control, stimulus presentation and behavior tracking were implemented using LabVIEW (National Instruments, Austin, Texas) and Matlab.

## Two-photon imaging

Functional 2-photon calcium imaging was carried out as described in *Antinucci et al., 2019* to record tectal activity in a single imaging plane in the absence of visual stimulus presentation for 30 min. Prior to imaging initiation, the larva was allowed to adapt to the imaging rig for 15 min.

## Calcium imaging analysis

All calcium imaging data analysis was performed using Matlab scripts. Volume motion correction was performed by 3D translation-based registration using the Matlab function 'imregtform', with gradient descent optimizer and mutual information as the image similarity metric. A registration template was generated as the time-average of the first 10 volumes and then iteratively updated following

each block of 10 newly registered volumes from the first 500 frames. This template was then used to register all remaining volumes. For elavl3:H2B-GCaMP6s experiments, 2D regions of interest (ROIs) corresponding to cell nuclei were computed from each template imaging plane using the cell detection code provided by *Kawashima et al., 2016*. For RGC axonal arbor imaging, two ROIs encompassing the tectal neuropil were manually defined for each imaging plane. The time-varying raw fluorescence signal $F_{raw}(t)$ for each ROI was extracted by computing the mean value of all pixels within the ROI mask at each time-point. A slowly varying baseline fluorescence $F_0(t)$ was estimated by taking the 10th percentile of a sliding 20 volume window, and was used to calculate the proportional change in fluorescence:

$$\frac{\Delta F}{F_0} = \frac{F(t) - F_0(t)}{F_0(t)}$$

These values were subsequently zero-centred by subtracting the mean for each ROI, and ROIs with a slow drift in their baseline fluorescence (for which the standard deviation of mean-normalised $F_0(t)$ was greater than 0.45) were discarded.

To estimate spike trains, the zero-centred $\frac{\Delta F}{F_0}$ time series for each cell was deconvolved using a thresholded OASIS (*Friedrich et al., 2017*) with a first-order autoregressive model (AR(1)) and an automatically-estimated transient decay time constant for each ROI.

To standardize the 3D coordinates of detected cell nuclei, template volumes were registered onto the *Tg(elavl3:H2B-RFP)* reference brain in the ZBB brain atlas (*Marquart et al., 2017*) using the ANTs toolbox version 2.1.0 (*Avants et al., 2011*) with affine and warp transformations. As an example, to register the 3D image volume in 'fish1_01.nrrd' to the reference brain 'ref.nrrd', the following parameters were used:

```
antsRegistration -d 3 -float 1 -o [fish1_, fish1_Warped.nii.gz] -n BSpline -r
[ref.nrrd, fish1_01.nrrd, 1] -t Rigid[0.1] -m GC[ref.nrrd, fish1_01.nrrd,
1, 32, Regular, 0.25] -c [200×200 × 200×0,1e-8, 10] -f 12×8 × 4×2 s 4×3 ×
2×1 t Affine[0.1] -m GC[ref.nrrd, fish1_01.nrrd, 1, 32, Regular, 0.25] -c
[200×200 × 200×0,1e-8,10] -f 12×8 × 4×2 s 4×3 × 2×1 t SyN[0.1,6,0] -m CC[
ref.nrrd, fish1_01.nrrd, 1, 2] -c [200×200 × 200x200×10,1e-7,10] -f 12×8 ×
4x2×1 s 4×3 × 2x1×0
```

Following registration, tectal ROIs were labeled using a manually created 3D mask (*Source data 1*).

## Burst detection

We used the same procedure to detect localised bursts of tectal activity in both recorded and simulated data. First, we found local maxima in tectal population activity after averaging the inferred spike trains of all tectal cells and smoothing with a sliding window (3 frames, 600ms). We next identified the population of cells that were active during a 1 s window centred on each peak and used the DBSCAN density-based clustering algorithm (*Ester et al., 1996*) to define spatial clusters based on Euclidean distances between the centroids of active cells. We excluded peaks during episodes of strong bilateral activation (commonly associated with vigorous swims or struggles) where more than 10% of the total tectal neurons were active and less than 70% of the active cells were located in the same hemisphere (16% ± 5% population peaks excluded per fish). We set the clustering distance threshold $\epsilon$ to 15 µm (~3 cell diameters) and the minimal number of cells in a neighbourhood, MinPts, to 12. MinPts was selected as the minimal number for which no clusters were detected after repeatedly performing circular permutations the time bases of all neurons, *Figure 1—figure supplement 1D–E*. After detecting the clusters associated with an activity peak, each was analysed independently to determine the initiation and termination times of the bursting activity of its constituent cells. A sliding window, six imaging frames in width (1.2 s) was evaluated at single frame increments. At each position, spike counts were discarded if they exceeded the 60[th] percentile of a Poisson cumulative distribution function with $\lambda$ equal to the mean number of inferred spikes across all cells in the cluster. Finally, the mean number of remaining spikes was computed and used to produce a vector describing the average Poisson-filtered firing within the cluster. Its first and last non-zero values were defined as the burst initiation and termination times.

## Activity correlation analysis

To assess the correlation structure of ongoing tectal activity, we calculated the Pearson correlation of the spiking activity of each neuron ('seed') and the activity of all other tectal neurons. Spiking vectors were smoothed by filtering with a Gaussian window with σ=1.4 frames (280 ms). To generate correlation maps, we first projected the spatial coordinates of tectal neurons onto two dimensions: anterior-posterior and medial-lateral tectal axes. This was done by manually tracing a curve along the tectal anterior-posterior axis and calculating the distance of each neuron to this curve (its medial-lateral location) and the location along the curve where it passes closest to each neuron (its anterior-posterior location). Next, the two-dimensional correlation map of each seed (excluding the seed itself, but including all other neurons) was fitted to a general bivariate normal distribution density function:

$$\frac{A}{2\pi\sigma_x\sigma_y\sqrt{1-\rho^2}}exp\left\{\frac{-1}{2(1-\rho^2)}\left[\left(\frac{x-\mu_x}{\sigma_x}\right)^2 + \left(\frac{y-\mu_y}{\sigma_y}\right)^2 - 2\rho\frac{(x-\mu_x)(y-\mu_y)}{\sigma_x\sigma_y}\right]\right\} + b$$

where $A$ is the gain, b is the bias, $\mu_x$ and $\sigma_x$ are the A-P axis mean and standard deviation, $\mu_y$ and $\sigma_y$ are the M-L axis mean and standard deviation, and $\rho$ is the correlation coefficient.

Alternatively, the correlation map was fitted to an exponential decay function:

$$A \cdot exp\left(-\sqrt{\left(\frac{x-\mu_x}{\sigma_x}\right)^2 + \left(\frac{y-\mu_y}{\sigma_y}\right)^2}\right) + b$$

A random location within 100 μm of the seed cell was used as an initial guess for the peak $\mu_x, \mu_y$. The following bounds were used when fitting:

$$b: [0,0.8], A: [0,inf], \mu_x: [-100,1200]\,\mu m, \mu_y: [-100,250]\,\mu m, \sigma_x: [0,500]\,\mu m, \sigma_y: [0,500]\,\mu m,$$

$$\rho: [-1,1]$$

## LNP spiking network model

The network model was based on the LNP stochastic spiking network formalism (*Pillow et al., 2005*; *Pillow et al., 2008*; *Truccolo et al., 2005*). Interactions between neurons (including feedback from a neuron onto itself) were modeled by temporal coupling filters and external (visual) input was considered instantaneous (see below).

The instantaneous firing rate of neuron j at time t, $\lambda_j(t)$, was derived by exponentiating its linear drive $\varphi(t)$, or sum of inputs:

$$\varphi(t) = x_j(t) + \Sigma_i\left[\left(I_{ij}^{(E)} - I_{ij}^{(I)}\right) \cdot y_i\right] + \mu \tag{1}$$

$$\lambda_j(t) = exp\,\varphi$$

where $x$ is the external stimulus, $I_{ij}^{(E)}$ and $I_{ij}^{(I)}$ are excitatory and inhibitory coupling filters, $y_i$ are spike train histories of each neuron at time $t$, and $\mu$ is a global baseline log-firing rate (DC input).

The excitatory and inhibitory coupling filters were each determined by three global parameters: gain (g), spatial influence standard deviation (σ) and temporal exponential decay time constant ($\tau$).

$$I_{ij}^{(X)}(t) = g^{(X)}exp\left(\frac{-d_{ij}^2}{2\sigma^{(X)2}} - \frac{t}{\tau^{(X)}}\right)$$

We also tested a model variant with exponential spatial decay function for connection weights (*Figure 3—figure supplement 1A*):

$$I_{ij}^{(X)}(t) = g^{(X)}exp\left(\frac{-d_{ij}}{\sigma^{(X)}} - \frac{t}{\tau^{(X)}}\right)$$

where X denotes excitatory (E) or inhibitory (I) interactions and $d_{ij}$ is the Euclidian distance between neuron $i$ and neuron $j$ in 3D standard coordinates following registration. If the two neurons reside in opposite tectal hemispheres, their coupling strengths were both scaled by 0.01.

In another model variant we eliminated 20% randomly chosen connections (both excitatory and inhibitory). To compensate for the reduced excitatory and inhibitory gain in the circuit we tuned the global gain parameters $(g^{(X)})$ accordingly.

The model simulation was implemented in Matlab (code available at https://github.com/azylbertal/TectalLNP copy archived at *Zylbertal, 2023*), based on code provided by *Pillow et al., 2008*. It uses a 50 ms timestep (resulting in 4:1 ratio with imaging rate) and comprises 14733 simulated neurons with locations inferred from the tecta of one fish. To significantly reduce memory and computational costs, a much longer (5 s) time step was used to simulate the slow inhibitory inputs to all cells. The inhibitory input was then interpolated before being summed with the excitatory input to produce the total linear drive. Spikes were drawn at random based on an inhomogeneous Poisson process with expected rates $\lambda_j(t)$, which were updated according to the equations above whenever at least one neuron spiked. To compare the simulation results with imaging results, simulated spiking output was binned to 200 ms windows, equivalent to imaging volumes.

## Model optimization

We tuned the seven model parameters $(g^{(E)}, g^{(I)}, \sigma^{(E)}, \sigma^{(I)}, \tau^{(E)}, \tau^{(I)}, \mu)$ to reproduce observed bursting characteristics using the Matlab implementation of controlled elitist evolutionary multi-objective optimization with a genotype-based crowding distance measure (*Kalyanmoy, 2001*). Specifically, we used three optimization objectives: (a) the log-rates of bursts binned according to the logarithm of the number of participating neurons; (b) the log-rates of bursts binned according to the logarithm of the burst duration; (c) mean burst-triggered spiking activity (specifically, the average activity of each neuron triggered on the peaks of bursts it participated in and subsequently averaged across all neurons, smoothed by spline interpolation and scaled from zero to one according to its 5th and 95th percentiles).

In each generation of the algorithm, a population of (initially random) 200 models was evaluated in parallel using UCL's Myriad computing cluster. Each simulation was run without external inputs for 306,000 steps (the equivalent of two hours and 15 min of imaging), and the initial 18000 steps (15 min) were discarded to avoid initialization transients. Localised bursts were detected based on the simulated spiking as they were detected for experimental data (see above). Since the inter-hemispheric interactions in the model are very weak (1% of the intra-hemispheric interactions), we were able to reduce computational cost during optimization by evaluating the models on one hemisphere. The loss value for each objective is the sum of squared errors between the simulation and experimental results. Parents for the next generation are selected by tournament: subsets of models are chosen by random (with replacement), and the best one from each set, that is the one with the highest ranking from the previous generation, is selected as a parent. 80% of the children are produced by intermediate crossover, where their parameters are chosen randomly from a uniform distribution bounded by the parameters of pairs of parents. The remaining 20% are produced by adaptive mutation, where they are randomly mutated with a direction that is maintained upon an increase in mean population fitness from the previous generation, or changed otherwise. The extended population (original population and children) is then evaluated and ranked based on Pareto front order, and within each front by genotype crowding distance (individuals with less close neighbors are ranked higher). Finally, the extended population is trimmed to retain the best 200 individuals for the next generation.

The evolution was stopped when the population scores stopped improving (~30 generations), producing a set of models on the approximated Pareto front – the solutions for which no individual model performs better than any other in all objectives. We chose the solution at the 'elbow' of the resulting front (the one closest to the origin in the z-scored loss space) and used it as a starting point for a local pattern search for final parameter refinement (*Figure 3—figure supplement 1D–E*).

## Network state estimation based on imaging data

We estimated the linear drive, or 'excitability' of each OT neuron prior to stimulus presentation (*Figure 4*) or spontaneous convergent saccade (*Figure 7*) by applying the spatiotemporal filters described by the LNP model to spiking history inferred from imaging data, regardless of whether other spontaneous events occurred during this time period (*Figure 4C*). For each event $i$, we calculated 60 s long vectors $\varphi_{ij}(t)$ representing the sum of inputs to each cell $j$, based on the activity of all other OT cells during the –60 to –5 s period prior to the event, weighted by time and distance per

the model parameters (*Equation 1* above). Since the model temporal filters assign larger weights to recent spikes, we ignored the spikes detected during the final 5 s to account for the inherent uncertainty in inferring spikes from calcium imaging data. To account for cell-specific biases, such as spatial edge effects, we corrected $\varphi_{ij}(t)$ by subtracting cell-specific 'baseline' mean linear drive vectors $\bar{\varphi}_j(t)$, estimated by repeating the calculation for 60 epochs of ongoing activity. The reported linear drive point-estimate was calculated by averaging the final 10 s of the baseline-corrected vectors. For each neuron, we calculated the percentage of variance in activity across presentations of a given stimulus type explained by the estimated linear drive according to a threshold-linear fit (*Figure 4D–E*):

$$y = \frac{a(x-x_0)+|a(x-x_0)|}{2} + c, a \geq 0$$

## Receptive field mapping and visual stimulus simulation

The simulated external input to each model neuron, $x_j(t)$, was estimated from the mean recorded activity of the corresponding tectal neuron as a function of stimulus azimuth ($\lambda_j(\alpha)$). To estimate receptive fields, the following function was fitted to this data:

$$\lambda_j^*(\alpha) = A_j exp\left(-\frac{(\alpha-\mu_j)^2}{2\sigma_j^2}\right)$$

producing values for gain ($A_j$), peak azimuth ($\mu_j$) and width ($\sigma_j$) for a specific neuron, stimulus elevation and motion direction (*Figure 5A–B*). To reduce the impact of noise, neurons with very low peak response or very narrow ($\sigma_j < 3°$) receptive fields were excluded from receiving external input in the model.

Finally, to infer external input drive from measured spiking output, we scaled the gain and receptive field widths (to account for apparent expansion caused by recurrent connectivity) to obtain:

$$x_j(t) = 2.5 \cdot 10^3 \cdot A_j exp\left(\frac{-(\alpha(t)-\mu_j)^2}{2(0.8\cdot\sigma_j)^2}\right)$$

Our results are robust to substantial variations in the choice of these scaling factors.

## Identification of responsive neurons

We identified tectal neurons as responsive to one of the stimulus elevations (*Figures 5–6*) if their mean response firing rate exceeded the 90th percentile over the responses of all neurons and was at least five times higher than their baseline firing rate.

## Statistical tests

p-values were obtained from two-tailed t-tests unless otherwise noted. For the two-way ANOVA used to model lateral activity bias (*Figure 7D*), post-saccadic gaze direction (rightward or leftward) and fish identity were used as independent factors, without interaction terms. The reported p-value is associated with the gaze direction.

## Acknowledgements

The authors thank members of the Bianco lab, James Fitzgerald, Eyal Wigderson and Avraham M Libster for helpful discussions and critical feedback on the manuscript and UCL Fish Facility staff for fish care and husbandry. AZ was supported by a BBSRC Discovery Fellowship (BB/S010564/1). IHB was supported by a Wellcome Trust and Royal Society Sir Henry Dale Fellowship (101195/Z/13/Z) and a Wellcome Trust Senior Research Fellowship (220273/Z/20/Z).

# Additional information

## Funding

| Funder | Grant reference number | Author |
|---|---|---|
| Wellcome Trust | 101195/Z/13/Z | Isaac H Bianco |
| Wellcome Trust | 220273/Z/20/Z | Isaac H Bianco |
| Biotechnology and Biological Sciences Research Council | BB/S010564/1 | Asaph Zylbertal |

The funders had no role in study design, data collection and interpretation, or the decision to submit the work for publication. For the purpose of Open Access, the authors have applied a CC BY public copyright license to any Author Accepted Manuscript version arising from this submission.

## Author contributions

Asaph Zylbertal, Conceptualization, Software, Formal analysis, Funding acquisition, Investigation, Visualization, Methodology, Writing - original draft, Writing - review and editing; Isaac H Bianco, Conceptualization, Resources, Software, Supervision, Funding acquisition, Methodology, Writing - original draft, Project administration, Writing - review and editing

## Author ORCIDs

Asaph Zylbertal ![ORCID] http://orcid.org/0000-0002-9789-2468
Isaac H Bianco ![ORCID] http://orcid.org/0000-0002-3149-4862

## Ethics

Animal handling and experimental procedures were approved by the UCL Animal Welfare Ethical Review Body and the UK Home Office under the Animal (Scientific Procedures) Act 1986, under Home Office project licence 70/8162 awarded to Isaac Bianco.

## Decision letter and Author response

Decision letter https://doi.org/10.7554/eLife.78381.sa1
Author response https://doi.org/10.7554/eLife.78381.sa2

# Additional files

## Supplementary files

• MDAR checklist

• Source data 1. Anatomical mask – tectal stratum periventriculare (SPV). TIFF stack containing binary mask defining the tectal SPV anatomical region, in ZBB space.

## Data availability

Neural activity and behavioural data have been deposited to Dryad, under https://doi.org/10.5061/dryad.q573n5tm5. Source data files have been provided for all figures. Simulation source code is available at https://github.com/azylbertal/TectalLNP (copy archived at *Zylbertal, 2023*).

The following dataset was generated:

| Author(s) | Year | Dataset title | Dataset URL | Database and Identifier |
|---|---|---|---|---|
| Zylbertal A, Bianco I | 2023 | Data for: Ongoing and visually evoked activity in the zebrafish optic tectum and adjacent brain structures | https://doi.org/10.5061/dryad.q573n5tm5 | Dryad Digital Repository, 10.5061/dryad.q573n5tm5 |

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
