## [Editor Report]

This important study investigates how neural activity states contribute to and shape sensory responses using a combination of neuronal activity imaging and computational modeling. They show that recurrent connectivity in networks can shape sensory responses in an experience-dependent manner and can be used to explain variability in experimentally-observed neuronal responses to sensory stimuli.

---

## [Decision Letter]

**Decision letter after peer review:**

Thank you for submitting your article "A recurrent network architecture explains tectal activity dynamics and experience-dependent behaviour" for consideration by *eLife*. Your article has been reviewed by 2 peer reviewers, and the evaluation has been overseen by a Reviewing Editor and Ronald Calabrese as the Senior Editor. The following individual involved in the review of your submission has agreed to reveal their identity: Benjamin Cowley (Reviewer #2).

Essential revisions:

1) Rewrite the manuscript to place a clear emphasis on the overall message of the manuscript.

2) Please include more model validations/comparisons (such as the effect of removing inhibition).

*Reviewer #1 (Recommendations for the authors):*

1. The abstract, introduction, and discussion need thorough revision so that the readers can understand the main take-home message. In its current shape, it is unclear whether the emphasis is on the utility of the LNP model or the proposal of slow, long-range inhibition between neurons in the tectum. I recommend emphasizing the latter aspect, given that the second half of this paper primarily focuses on its role. I agree that they show an excellent network model, but this is still a hypothetical product and the authors did not perform a thorough comparison with other types of models which may fit equally well.

2. I advise the authors to revise the last paragraph of Page 20 and the next one in the Discussion section that starts from "However, this study calls these assumptions into question." I agree that the author's LNP model explains some critical aspects of tectal bursting dynamics (that are defined by the authors), but they did not perform any analyses or model comparison to exclude previously proposed nonuniformities in the connections between neurons. This is an unfair argument and likely provokes unneeded drawbacks from colleagues in the field.

Also, there are existing results and unpublished studies showing that there are heterogeneities in neurotransmitter types in the tectal population that are nonuniformly distributed. Therefore, I advise the authors to rewrite this section to discuss how the potential uniform connectivities between neurons, which are probably shaped through proximity-based mechanisms, co-exist with connectivities that are shaped by other mechanisms to support diverse functionalities in the optic tectum.

3. I advise the authors to dig into the property of the LNP model a bit more to emphasize the importance of hypothetical inhibitory connectivities. It would help the reader to see, for example, how different temporal and distance coefficients of inhibitory connections will alter the nature of spontaneous bursts. Figure 3 uses a space to explain how EMOO process works, but this is distracting and should be moved to the supplementary figure.

4. I recommend having a main figure panel that explains how the authors calculate the "linear drive". It is written in the methods, but this "linear drive" measure is everywhere in the manuscript and it is better to have a dedicated panel.

*Reviewer #2 (Recommendations for the authors):*

Overall, I enjoyed the paper, and I think it will be a nice addition to the community trying to model trial-to-trial neural variability. In the public review, I basically veiled the analyses I think would solidify the work. Namely, making sure the model is thoroughly evaluated, shoring up some questions I had about the data results, and perhaps linking bursting to the stimulus-evoked activity analyses in some way. I think the claim that a uniform recurrent connectivity motif can also explain bursting (versus assemblies) is fine, but it is not as strong as "better explains the data than an assemblies model". Right now, the reader is left with the question that different models can explain bursting but no definitive answer on which one is correct. If there is a formally-proposed assemblies model, a model comparison would address this issue. More importantly, compare the gaussian distance model with other possible fall-offs (such as other radial basis functions and as a baseline – a uniform model with no distance fall-off) and check the assumption that each neuron falls off at the same rate (i.e., compare σ_x_ and σ_y_ across neurons). Considering Pillow et al., 2008 was an inspiration for the model, reporting prediction performance R^2^ (or log-likelihood as in Pillow 2008) for different models (e.g., with/without an inhibitory gain) would provide strong evidence for your claims. Fitting the parameters of a GLM and also showing this gaussian fall-off would show strong support. A key difference between a GLM and the distance model is that a GLM can take into account correlations between the neighboring neurons whereas the distance model assumes all neighboring neurons are independent. In other words, linear regression may take advantage of correlations in input features X (e.g., by subtracting one neuron's activity from another), but the distance model cannot (and thus may not be able to remove spurious noisy signals).

The reported R^2^s (Figure 4D and 7H, blue distributions) are really weak (R^2^ ~ = 0.05 for most). Please come up with a null R^2^ distribution (e.g., reverse time, flip signs) and compare the actual R^2^ distribution to the null. This ensures there is some signal (even if tiny).

[Editors’ note: further revisions were suggested prior to acceptance, as described below. Please see the Author Response section below for the authors responses to these requests. ]

Thank you for resubmitting your work entitled "Recurrent network interactions explain tectal response variability and experience-dependent behaviour" for further consideration by *eLife*. Your revised article has been evaluated by Timothy Behrens (Senior Editor) and a Reviewing Editor.

The manuscript has been improved but there are some remaining issues that need to be addressed, as outlined below:

The revised version seems to have addressed many of the points raised by the reviewers, yet no new modeling has been done as suggested in the original reviews. Some points that were raised during the consultation were:

"…some basic ablations/model comparisons would go a long way. E.g., how useful is the surround inhibition? What if you were to fit σ_E_ and σ_I_ for each ROI (versus assuming the same σ_E_ and σ_I_ across ROIs)? If computation time is a problem, they can always subsample. As it stands, their model of "linear drive" explains ~1-2% of the variability (Figure 4e)…which suggests something is missing."

"I did not see the authors' serious efforts in improving the rigor of their LNP model and its comparison with other models. We pointed them out as major concerns, and some of them are not so difficult to address. "

Please take these comments into consideration while preparing the revision.

*Reviewer #2 (Recommendations for the authors):*

I accept this manuscript for publication. I have read the rebuttal and reviewed the manuscript. Although the authors did not try any further modeling as recommended by this reviewer, this simple model may be of use to the zebrafish field as an instantiation of "nearby neurons do similar things". The low performance in predicting residuals (Figure 4E, < 2% explained variance) suggests there is much room for improvement in modeling these interactions.

---

## [Author Response]

Essential revisions:1) Rewrite the manuscript to place a clear emphasis on the overall message of the manuscript.

We agree with the reviewers that the main message was not sufficiently clear in the original manuscript. We have now completely rewritten the Abstract and Introduction and made substantial changes within the Results and Discussion to address this. Within the context of understanding response variability, we focus on the role for activity-dependent changes in network state. We highlight that the utility of the model is that it enables us to estimate the (unseen) excitability state of tectal neurons, based on the recent history of spiking activity, and thereby explain a fraction of trial-to-trial variability in visually evoked activity, response adaptation and behavioural habituation. In the Introduction and Results, we have tried to reinforce the links between the different parts of the paper. For instance, we now emphasise that examining ongoing activity allowed us to parameterise spatiotemporal interactions in OT and this in turn gave us the means to estimate the excitability state of biological neurons based on recent recorded activity (and thereby assess how incidental network state accounts for visual response variability). We hope that these changes have been successful in presenting the study as a coherent whole.

2) Please include more model validations/comparisons (such as the effect of removing inhibition).

We have added extensive additional exploration and validation of the model in line with the reviewer comments (further details below). In brief, we ran the simulation under a broader range of parameter values (including lack of inhibition and distance-independent connectivity), tested an alternative function for the spatial dependency of intercellular interactions and assessed robustness by randomly removing a subset of the connections. We have also collected new experimental data and performed a more statistically robust comparison against model predictions (in particular for figures 4 and 7).

Reviewer #1 (Recommendations for the authors):1. The abstract, introduction, and discussion need thorough revision so that the readers can understand the main take-home message. In its current shape, it is unclear whether the emphasis is on the utility of the LNP model or the proposal of slow, long-range inhibition between neurons in the tectum. I recommend emphasizing the latter aspect, given that the second half of this paper primarily focuses on its role. I agree that they show an excellent network model, but this is still a hypothetical product and the authors did not perform a thorough comparison with other types of models which may fit equally well.

We hope we have addressed this important comment in this revised version. We focus on the goal of understanding variability in neural activity and behaviour and specifically how recent network activity can cause changes in brain state that account for (a fraction of) this variability. We emphasise our finding of an important role for long-lasting activity-dependent suppression in OT and explain that a key utility of the model was that it allowed us to estimate network state from recently observed activity.

2. I advise the authors to revise the last paragraph of Page 20 and the next one in the Discussion section that starts from "However, this study calls these assumptions into question." I agree that the author's LNP model explains some critical aspects of tectal bursting dynamics (that are defined by the authors), but they did not perform any analyses or model comparison to exclude previously proposed nonuniformities in the connections between neurons. This is an unfair argument and likely provokes unneeded drawbacks from colleagues in the field.Also, there are existing results and unpublished studies showing that there are heterogeneities in neurotransmitter types in the tectal population that are nonuniformly distributed. Therefore, I advise the authors to rewrite this section to discuss how the potential uniform connectivities between neurons, which are probably shaped through proximity-based mechanisms, co-exist with connectivities that are shaped by other mechanisms to support diverse functionalities in the optic tectum.

We have revised this discussion. We certainly did not intend to claim that OT is uniform nor that subnetworks (assemblies) do not exist. Rather, we want to highlight that bursting is not, in and of itself, sufficient evidence because our simplified model with a uniform recurrent connectivity rule can generate busting with biologically plausible statistics. As the reviewer notes, the tectum has complex circuitry and we hope that future studies will be able to directly measure connectivity to assess how it shapes tectal dynamics and computations.

3. I advise the authors to dig into the property of the LNP model a bit more to emphasize the importance of hypothetical inhibitory connectivities. It would help the reader to see, for example, how different temporal and distance coefficients of inhibitory connections will alter the nature of spontaneous bursts. Figure 3 uses a space to explain how EMOO process works, but this is distracting and should be moved to the supplementary figure.

We have significantly expanded the exploration of model parameter space, with emphasis on the inhibition parameters as suggested by the reviewer (Figure 3A and associated text). This includes a broad space of inhibition strengths (including a complete lack of inhibition where bursting ceases due to unopposed excitation) and a broad scale of space constants (extending to the limiting case where connectivity is independent of distance). As suggested, we have moved the EMOO schematic to supplement.

4. I recommend having a main figure panel that explains how the authors calculate the "linear drive". It is written in the methods, but this "linear drive" measure is everywhere in the manuscript and it is better to have a dedicated panel.

We have added a figure panel to explain how it is calculated from observed spiking (Figure 4C).

Reviewer #2 (Recommendations for the authors):Overall, I enjoyed the paper, and I think it will be a nice addition to the community trying to model trial-to-trial neural variability. In the public review, I basically veiled the analyses I think would solidify the work. Namely, making sure the model is thoroughly evaluated, shoring up some questions I had about the data results, and perhaps linking bursting to the stimulus-evoked activity analyses in some way. I think the claim that a uniform recurrent connectivity motif can also explain bursting (versus assemblies) is fine, but it is not as strong as "better explains the data than an assemblies model". Right now, the reader is left with the question that different models can explain bursting but no definitive answer on which one is correct. If there is a formally-proposed assemblies model, a model comparison would address this issue.

As we discuss above, we developed an intentionally simple model to capture the principal intercellular interactions by which activity is fed back to modulate the excitability state of tectal neurons. We did not attempt to fit cell-specific connectivity profiles and did not seek to engineer assemblies (which would vastly expand the parameter space and inevitably support bursting). Our results do not rule out assemblies, but rather we demonstrate that their existence is not a prerequisite for localised bursting. Assemblies might well exist, but future studies will need to find direct evidence for them (and see response to a comment on this subject by reviewer 1).

More importantly, compare the gaussian distance model with other possible fall-offs (such as other radial basis functions and as a baseline – a uniform model with no distance fall-off) and check the assumption that each neuron falls off at the same rate (i.e., compare σ_x_ and σ_y_ across neurons).

See responses above.

Considering Pillow et al., 2008 was an inspiration for the model, reporting prediction performance R^2^ (or log-likelihood as in Pillow 2008) for different models (e.g., with/without an inhibitory gain) would provide strong evidence for your claims. Fitting the parameters of a GLM and also showing this gaussian fall-off would show strong support. A key difference between a GLM and the distance model is that a GLM can take into account correlations between the neighboring neurons whereas the distance model assumes all neighboring neurons are independent. In other words, linear regression may take advantage of correlations in input features X (e.g., by subtracting one neuron's activity from another), but the distance model cannot (and thus may not be able to remove spurious noisy signals).

See above concerning our use of the model to simulate activity rather than fit recorded data and the extensions to parameter space (including no inhibitory gain). Note that in the model, recurrent connectivity ensures that the activity of an individual cell is strongly influenced by other neurons in its vicinity.

The reported R^2^s (Figure 4D and 7H, blue distributions) are really weak (R^2^ ~ = 0.05 for most). Please come up with a null R^2^ distribution (e.g., reverse time, flip signs) and compare the actual R^2^ distribution to the null. This ensures there is some signal (even if tiny).

This is addressed by the null distributions in the revised analysis for figures 4 and 7 (see above).

[Editors’ note: further revisions were suggested prior to acceptance, as described below.]

Reviewer #2 (Recommendations for the authors):I accept this manuscript for publication. I have read the rebuttal and reviewed the manuscript. Although the authors did not try any further modeling as recommended by this reviewer, this simple model may be of use to the zebrafish field as an instantiation of "nearby neurons do similar things". The low performance in predicting residuals (Figure 4E, < 2% explained variance) suggests there is much room for improvement in modeling these interactions.

In the newly revised version, we have extensively overhauled the analysis of visual response variability (Figure 4). In short, we now focus on single-cell trial-to-trial variability and find that model-predicted linear drive explains on average ~20% variance (i.e. an order of magnitude improvement).

In the original analysis, we used linear regression to try to explain, for individual trials, the response residuals *across cells* in terms of their model-predicted linear drive. While for some trials this could explain a substantial fraction of the variance (achieving r^2^>0.2), it was less successful for other trials, resulting in the low median r^2^ flagged by the reviewer. When we inspected trials for which predictive performance was poor, we discovered a common cause was that there was little variance either in LD or response residuals across cells (for example when OT is broadly suppressed), inevitably resulting in a low r^2^.

The current analysis resolves this issue because we have explicitly focussed on *trial-to-trial variability* of visual responses *for individual cells*. Specifically, for each visually responsive neuron in turn (defined as cells with a non-zero median response across stimulus presentations), we explain visual responses across trials in terms of the pre-stimulus model-predicted linear drive for those same trials. We fitted a rectified-linear function since a negative spiking response is not possible. This results in a median r^2^ of 0.2 (average across fish). For comparison, we again include estimates using linear drive from a null model (random times, r^2^ ~0.02) and the baseline model (~0.1). It should be noted that the baseline model is expected to do significantly better than the null model, as it shares the basic architecture with the optimized model, albeit with different time and space constants.

We consider this revised analysis is easier to understand, aligns better with the motivation of the study (seeking to understand how activity shapes internal state and thence response variability *across trials*), as well as having superior predictive performance.